# Quantifying Variations in Shortwave Aerosol-Warm Cloud-Radiation Interactions Using Local Meteorology and Cloud State Constraints

Alyson Douglas[1] and Tristan L'Ecuyer[1]

[1]University of Wisconsin-Madison

**Correspondence:** Alyson Douglas (ADouglas2@wisc.edu)

**Abstract.** While many studies have tried to quantify the sign and the magnitude of the warm marine cloud response to aerosol loading, both remain uncertain owing to the multitude of factors that modulate microphysical and thermodynamic processes within the cloud. Constraining aerosol-cloud interactions using the local meteorology and cloud liquid water may offer a way to account for covarying influences, potentially increasing our confidence in observational estimates of warm cloud indirect effects. Four years of collocated satellite observations from the NASA A-Train constellation, combined with reanalaysis from MERRA-2, are used to partition marine warm clouds into regimes based on stability, the free atmospheric relative humidity, and liquid water path. Organizing the sizable number of satellite observations into regimes is shown to minimize the covariance between the environment or liquid water path and the indirect effect. Controlling for local meteorology and cloud state mitigates artificial signals and reveals substantial variance in both the sign and magnitude of the cloud radiative response, including regions where clouds become systematically darker with increased aerosol concentration in dry, unstable environments. A darkening effect is evident even under the most stringent of constraints. These results suggest it is not meaningful to report a single global sensitivity of cloud radiative effect to aerosol. To the contrary, we find the sensitivity can range from -.46 to .11 $\frac{\text{W m}^{-2}}{\ln(\text{AI})}$ regionally.

## 1 Introduction

Warm clouds play an important role in Earth's radiative balance. Cooling the atmosphere and covering 25% of the Earth's surface on average and reflecting incoming shortwave radiation, any changes to their radiative properties should be well quantified and understood (Hahn and Warren, 2007). These clouds are most prevalent off the western coasts of continents as marine stratocumulus, near the tropics as trade cumulus, and in the storm track regions as stratus (Ackerman et al., 2018). Perturbations in aerosol, whether from natural sources like sea spray or anthropogenic activities like biomass burning, lead to cloud-aerosol interactions that alter cloud radiative properties through two main effects, the albedo and the cloud lifetime effects. First termed by Twomey in 1977, the albedo effect, or the first indirect effect as it's also known, suggests that clouds will become brighter as a result of aerosol loading. For a fixed liquid water path, increased aerosol within a cloud increases the number of cloud condensation nuclei (CCN), forcing the mean drop size to decrease, resulting in a brighter, more reflective cloud. The second indirect effect, or the cloud lifetime effect, proposed by Alrecht (1989) builds on this idea, noting that a decrease in mean

drop size due to aerosol-cloud interactions may also delay the onset of collision coalescence, suppressing precipitation and, in turn, allowing the cloud to survive longer, grow larger, and ultimately reflect more shortwave radiation. Early estimates of the indirect effect estimated including the cloud lifetime effect may increase it by 1.25x (Penner et al., 2001). Work since then has concentrated on decreasing the range of uncertainty rather than separating the effects in observation based studies, as without

explicit constraints in place on the cloud water, the two effects are intrinsically related through the liquid water content of the cloud (Mülmenstädt and Feingold, 2018).

However, observing the indirect effect is not as straight forward as looking out your window trying to spot brighter clouds. The magnitude and sign of the indirect effect is extremely sensitive to the method used to quantify it. As a result, the Inter-governmental Panel on Climate Change (IPCC) has low confidence in the current estimate of the global aerosol indirect effect

(AIE) (Boucher et al., 2013). An accurate assessment of the total indirect effect will reduce error in climate sensitivity and further our understanding of the role of clouds in future climates (Bony and Dufresne, 2005).

Historically, methods of estimating the AIE employ a single linear regression of either the cloud's radiative effect or droplet radius against a proxy for aerosol concentration (Platnick and Twomey, 1994; Lohmann and Feichter, 2005; Christensen et al., 2016). This method ignores all possible covariances between the cloud, aerosol, and any processes that may effect both and

assumes one linear regression captures all effects, disregarding the role played by the local environment as a strong modulator of warm cloud properties and responses (Stevens and Feingold, 2009). Constraining the local meteorology, or the characteristics of the environment around the cloud, as well as cloud type can significantly alter the magnitude of the AIE compared to single, unconstrained global linear regression (Gryspeerdt et al., 2014). Regional analyses, such as treating the marine stratocumulus cloud decks off the west coasts as a homogeneous sample, instead capture assorted responses and magnitudes as they fail to

extricate covariance with local meteorology (Bender et al., 2016). Observationally-based estimates simply cannot "turn off" the effects of entrainment or other environmental effects like a model, therefore observation based approaches must prescribe a way to diminish the effect of these influences on cloud radiative effects, even at a regional scale.

Modeling provides one pathway for estimating the global AIE that explicitly accounts for local meteorological conditions, however low clouds are one of the largest sources of error in current global climate models (GCM) (Williams and Webb, 2009).

In particular, GCMs tend to overestimate liquid water path (LWP) in low clouds, which leads to an overestimation of the albedo (Nam et al., 2012). The artificially elevated LWP impacts the sensitivity to aerosol by assessing it under unrealistic conditions. Further, entrainment and precipitation are artificially dampened as a result of incorrect cloud parameterizations in GCMs (Tsushima et al., 2016; Lee et al., 2009). Many cloud-aerosol processes are explicitly resolved in large eddy simulation (LES) models, but these are limited to small scales. LES can prescribe exact environments, but again these are limited to idealized

meteorologies, only realistic to small regions on Earth. The microphysical processes of aerosol activation, nucleation, and eventual raindrop formation can only be parameterized in current GCMs and will remain so for the foreseeable future. The resolution is too coarse to emulate all scales of aerosol-cloud interactions hence the dependence on parameterizations and large uncertainty in model-derived estimates (Wood et al., 2016). A solution to this problem is a combination of global climate modeling guided by observation-based analysis and coordinated LES modeling to understand and quantify the AIE (Stephens,

2005).

Observation-based methods must avoid the pitfalls of historical evaluations and define a clear methodology to limit covariance with local environmental conditions or buffering by the cloud. Buffering is when the cloud state and/or environment work to reduce the impact of aerosols on the cloud Stevens and Feingold (2009). Cloud characteristics, such as LWP, and the local meteorology, like stability, can compound uncertainty in evaluating the AIE because they influence both radiative properties and susceptibility to aerosol (Lee et al., 2009; Feingold and Kreidenweis, 2002). The AIE is specifically defined as the cloud response to aerosol and the resulting effects on the radiative properties. Any quantification of the AIE must avoid including the effects of the local environment on the cloud radiative properties. When the local meteorology was accounted for, Gryspeerdt et al. (2016) found the sensitivity of cloud fraction to aerosol loading was reduced by 80%. Quantifying the AIE therefore requires separating and constraining all processes that moderate cloud radiative properties from those specifically due to aerosol-cloud radiation interactions (Stevens and Feingold, 2009). Organizing clouds into constrained, bounded spaces based on the external and internal covarying conditions can improve aerosol-cloud-radiation impact estimates (Ghan et al., 2016).

This study examines the sensitivity of the shortwave radiative forcing of warm clouds to aerosol by employing a methodology that attempts to adequately constrain external influences while maintaining sufficiently robust statistics. Our methodology takes advantage of the vast sampling provided by satellites to systematically hold environmental conditions and cloud state approximately constant. We quantify the warm cloud sensitivity to aerosol for clouds of similar properties within similar environments. While most satellite studies of aerosol-cloud interactions are by necessity correlative, the more covarying factors that are controlled (at the individual cloud level), the more closely we can approximate a causal relationship. Although we cannot confirm causation due to the temporal resolution of the observations, some studies have begun utilizing the high temporal resolution of geostationary satellites to augment A-Train observations and fix this ongoing problem (Sauter and L'Ecuyer, 2017). In our study, a set of environmental conditions and cloud state parameters is referred to as a regime. This idea of stratifying observations into regimes has been successfully implemented before to analyze cloud processes (Williams and Webb, 2009; Chen et al., 2014; Gryspeerdt and Stier, 2012; Oreopoulos et al., 2016).

The environmental and cloud state regimes adapted here are designed to homogenize the clouds and processes occurring, reducing covariance the cloud radiative response to aerosol and other influences. Observationally-based, regime-dependent cloud processes have been discerned most often over large regional scales, however, divergent signals can be lost depending on the size of the region analyzed (Grandey and Stier, 2010). Even on small, local scales, variance in the meteorology alters the strength of the observed effects (Liu et al., 2016). A study using satellite observations with regime constraints, for example, found a definite relationship between the warm cloud AIE varies and atmospheric stability on a global scale (Chen et al., 2014).

One important meteorological influence is the stability of the boundary layer. LES of warm clouds have further shown that environmental instability can alter the effects of aerosol loading on warm clouds (Lee et al., 2012). The need to incorporate stability into AIE estimates has also been noted in prior observational studies (Sorooshian et al., 2009; L'Ecuyer et al., 2009; Su et al., 2010). Warm clouds in stable environments may show an increasing LWP with respect to aerosol loading while unstable environments may exhibit a decrease in LWP (Chen et al., 2014). Su et al. (2010) found the stability and rate of subsidence work to modulate aerosol-cloud-radiation interactions in warm clouds.

The effects of large scale subsidence and entrainment can be captured by the relative humidity ($RH_{700}$) in the free atmosphere, known to exert a powerful influence on warm cloud characteristics (Wood and Bretherton, 2004). Entrainment of free atmospheric air furthers the decoupling process by increasing the temperature and humidity gradients at the cloud top (Lewellen and Lewellen, 2002). Including $RH_{700}$ in aerosol sensitivity studies accounts for some decoupling influence. Models affirm the effects of entrainment on the cloud layer depend in part on $RH_{700}$, as LES have shown RH differences moderates cloud feedbacks in low warm cloud simulations (Van der Dussen et al., 2015). De Roode et al. (2014) showed that $RH_700$ plays a significant role in modulating the liquid water path, which could then modulate the strength of any aerosol-cloud interactions. This modulation is likely due to the entrainment of dry air from the free atmosphere which alters the distribution of liquid water within a cloud (Ackerman et al., 2004; Bretherton et al., 2007).

In his original work, Twomey postulated that cloud albedo ought to increase with aerosol provided LWP is held fixed, after all, albedo is dependent on the optical depth and effective radius. The LWP has been shown to clearly control the second AIE via its influence on precipitation suppression (L'Ecuyer et al., 2009; Sorooshian et al., 2009). Field campaign observations have noted this relationship as well. For example, the Atmospheric Radiation Measurement Mobile Facility Azores campaign fond the cloud radiative response depended largely on the LWP (Liu et al., 2016). LWP is intrinsically tied to the magnitude of the AIE . Failing to distinguish clouds by LWP will lead to large covariance and/or buffering in the system by the LWP.

For these reasons, we adopt the boundary layer stability and relative humidity of the free atmosphere in conjunction with LWP to segment observations into regimes at the individual satellite pixel scale. To illustrate the impact of these specific buffering factors, we sequentially increase constraints on the regression of the warm cloud radiative effect against aerosol, what we refer to as the sensitivity or $\lambda$. First, the sensitivity is constrained by only LWP to demonstrate the importance of accounting for cloud state alone when estimating aerosol response. Next, environmental regimes of stability and relative humidity are used segment warm clouds and, within each regime, the sensitivity of the cloud radiative effect to aerosol is assessed. These environmentally regimented observations are then further separated into LWP regimes to control for cloud state and environment simultaneously. Finally, the warm cloud sensitivity with all regime constraints is derived on a regional basis to account for local influences not captured by the global regime partitions.

## 2 Methods

### 2.1 Data

The effect of aerosol on marine warm cloud shortwave radiative properties is diagnosed from observations collected by the NASA A-Train constellation from 2007 to 2010. The A-Train is a series of synchronized satellites which allow for collocated observations from a variety of instruments (L'Ecuyer and Jiang, 2011). Environmental information is provided by collocated reanalysis data from the Modern-Era Retrospective analysis for Research and Applications Version 2 (MERRA-2). Collocated observations from multiple instruments, combined with high resolution reanalysis at the pixel scale, allows an extensive vie of the roles the environment and cloud state play in modulating the warm cloud sensitivity to aerosol concentration.

## 2.2 Cloud

The Cloud Profiling Radar (CPR) on CloudSat and the Cloud-Aerosol Lidar and Infrared Pathfinder Satellite (CALIPSO) are used to restrict analysis to single-layer, marine warm clouds between 60° N and 60° S. All data is interpolated down to CloudSat's ∼ 1km footprint. The CloudSat 2B-CldClass-Lidar product that classifies cloudy pixels based on their vertical structure from merged radar and lidar observations is leveraged to filter out ice phase and multilayered cloud systems (Sassen et al., 2008; Austin et al., 2009). All observations are restricted to below the freezing level of CloudSat which is determined using an ECMWF-AUX collocated reanalysis dataset and set where ECMWF determines the 0° isotherm. The Advanced Microwave Scanning Radiometer - Earth Observing System (AMSR-E) liquid water path (LWP) aboard the Aqua satellite is then used to limit observations to scenes where the LWP is above .02 $\frac{kg}{m^2}$ and below .4 $\frac{kg}{m^2}$ (Wentz and Meissner, 2007). Very thin clouds below .02 $\frac{kg}{m^2}$ are likely thin veil clouds with low albedos that are not the focus of this analysis (Wood et al., 2018). An along-satellite track cloud fraction is determined by finding the average number of warm cloud pixels that satisfy these criteria (seen by CloudSat or CALIPSO, below the CloudSat determined freezing level, and LWP between .02 and .4 $\frac{kg}{m^2}$) over each 12 km segment of the CloudSat track on a pixel by pixel basis, a scale that represents both the local scale length of the boundary layer and field-of-view used to define cloud radiative effects from Clouds and the Earth's Radiant Energy System (CERES) (Oke, 2002). Marine warm clouds fitting these parameters reside within the boundary layer. Even with these initial constraints on LWP and height, there were 1.8 million satellite observations fitting these parameters within the time period.

The warm cloud shortwave radiative effect is found by combining this along track warm cloud fraction with top of atmosphere (TOA) radiative fluxes from CERES. CERES has a total (.4 - 200 $\mu m$) and shortwave channel (0.4 - 4.5 $\mu m$) that allow outgoing shortwave and longwave fluxes at the top of the atmosphere to be estimated using appropriate bi-directional reflectance models. All-sky radiances from CERES are not restricted to any type of scene and include the raw radiances observed by CERES. The shortwave warm cloud radiative effect (CRE) is then defined in terms of the all sky and inferred clear sky forcings from CERES and warm cloud fraction from CloudSat. The clear sky flux ($F^{\uparrow}_{Clear\ Sky}$) is a regional, monthly mean estimate of cloud free outgoing shortwave radiation. Writing the all-sky net SW radiation at the top of the atmosphere as:

$$(F^{\downarrow}_{SW} - F^{\uparrow}_{SW})_{All\ Sky} = (F^{\downarrow}_{SW} - F^{\uparrow}_{SW})_{Clear\ Sky} \times (1 - CF) + (F^{\downarrow}_{SW} - F^{\uparrow}_{SW})_{Cloudy} \times CF \tag{1}$$

It is easy to show that for shortwave radiances:

$$F^{\uparrow}_{All\ Sky} - F^{\uparrow}_{Clear\ Sky} \times (1 - CF) = CRE \tag{2}$$

where the warm CRE$_{SW}$ = CF $\times$ $F^{\uparrow}_{Cloudy}$

The instantaneous CRE for each warm cloud observation is used in conjunction with aerosol information and corresponding instantaneous cloud state and meteorological state constraints to derive the sensitivity of the cloud radiative effect to aerosol loading.

## 2.3 Aerosol

Aerosol index (AI) is used to characterize the concentration of aerosol in the atmosphere. AI is the product of the Angstrom exponent (found using AOD at 550 and 870 nm) and AOD at 550 nm, both of which are derived from the Moderate-Resolution Imaging Spectroradiometer (MODIS) aboard the Aqua satellite. The Angstrom exponent, a measure of the turbidity of the atmosphere, is derived from multiple estimates of aerosol optical depth (AOD) (Ångström, 1929; Remer et al., 2005). The MODIS Angstrom exponent provides information about the size of the observed aerosol as well as concentration (Levy et al., 2010). MODIS AI is derived from the auxiliary dataset (MOD06-1km-AUX) developed from the overlap of the CloudSat CPR footprint and the MODIS cloud mask at pixel level. Although AI is not a direct measurement of CCN in the air, it has a higher correlation with CCN compared to the AOD and is therefore more suitable for aerosol-cloud interaction studies (Stier, 2016; Dagan et al., 2017). While AOD and the Angstrom exponent from MODIS are not available in cloud scenes, the collocated dataset interpolates these between clear sky scenes in order to infer a cloudy AI. For lower cloud fraction scenes, this interpolation is more accurate, however it is possible that in higher cloud fraction scenes, the accuracy of AI is reduced. This is a source of uncertainty within our results. AI can be affected by aerosol swelling in the most humid environments. All results have some amount of uncertainty due to this effect (Loeb and Schuster, 2008). This is minimized in the driest $RH_{700}$ regimes, however the most humid $RH_{700}$ regimes may be affected by aerosol swelling. The effect will be largest in the cloudiest regions such as the marine stratocumulus decks in the South Atlantic, Southeast Pacific, and off the California coast because aerosol measurements near clouds ( 15 km) are subjected to the largest amount of swelling (Christensen et al., 2017). It has been suggested using AI underestimates the strength of the indirect effect; our estimates of sensitivity of the warm cloud radiative effect to aerosol could be thought of a lower bound on the warm cloud indirect effect sensitivity (Penner et al., 2011). Another source of uncertainty is that the aerosol may not be located at the same height as the warm, boundary layer clouds we are evaluating. Aerosol should ideally be located near the cloud base in order to be fully activated and initiate the indirect effect (Chen et al., 2018).

## 2.4 Regimes

### 2.4.1 Environmental Regimes

MERRA-2 reanalyses collocated with each CloudSat footprint is used to define local thermodynamic conditions that distinguish environmental regimes. The environmental regimes employed here provide a crude representation of the local meteorology acting to inhibit or invigorate the cloud response. While these states, defined from percentile bins of the estimated inversion strength (EIS) and relative humidity at 700 mb ($RH_{700}$), do not capture the complete range of environmental factors that influence warm cloud development, they have been shown to provide fairly robust bulk classification for sorting satellite observations into meteorological regimes (Sorooshian et al., 2009; L'Ecuyer et al., 2009; Chen et al., 2014). Here, EIS is calculated using MERRA-2 temperature and relative humidity profiles and indicates the stability of the boundary layer. EIS incorporates effects of water vapor on the lower tropospheric static stability and is better correlated for all cloud types with cloud fraction.

From Wood and Bretheron (2006):

$$EIS = LTS - \Gamma_m^{850}(z_{700} - LCL) \tag{3}$$

where $\Gamma_m^{850}$ is the moist-adiabatic potential temperature gradient and LTS is the lower-tropospheric stability.

The relative humidity at 700 mb is used as a measure of the effect of entraining free tropospheric air (Karlsson et al., 2010). As the height of the 700 mb isobar is included in the equation for EIS, there is some covariability between EIS and $RH_{700}$. Some processes involved in altering the height at 700 mb will also affect $RH_{700}$ and vice versa, therefore there is some covariability between our two meteorological variables. When referring to the effects of entrainment, it means the effects of $RH_{700}$. All observations within the 5% - 95% percentiles of both EIS and $RH_{700}$ are partitioned into regimes of percentile limits. The bin limits depend on the number of bins implemented, which is varied in the results to establish the degree to which the environment must be constrained to accurately characterize sensitivity. For example, with 100 environmental regimes, the observations will be binned from by 10 percentile limits of both EIS and $RH_{700}$. Within each row of $RH_{700}$ of the environmental regimes, there are the same number of observations as within each column of EIS, however, within each individual environmental regime of both EIS and $RH_{700}$, the number of observations is dependent on the distribution of both EIS and $RH_{700}$.

### 2.4.2 Cloud States

Cloud states are defined by the LWP. Although there are other definitions of cloud regimes and cloud states used in other studies (e.g. Oreopoulos et al. (2017)), throughout ours cloud state or cloud morphology refers to the set of observations binned by liquid water path. Environmental stability and entrainment directly affect the LWP so these parameters are not independent. In what follows, however, we consider the LWP separately from the local meteorology to separately evaluate two aspects of the indirect effect formulation. Since Twomey's original hypothesis of the aerosol indirect effect was based on holding LWP constant, we first examine the impact of increasing stringent constraints on LWP. Constraining LWP diminishes the effects of aerosol on cloud LWP itself allowing the sensitivity of the warm cloud CRE to aerosol to be isolated (Gryspeerdt et al.). More recently, numerous others have extensively demonstrated that aerosol indirect effects can be buffered by other environmental conditions. Since EIS and RH have been frequently adopted as proxies for these buffering effects, we further examine the impact of increasingly stringent constraints on these environmental characteristics. Our separation of 'cloud regimes' and 'meteorological regimes' is made only to contrast the magnitudes of their effects and does not imply that LWP is independent of EIS or RH. Ultimately it will be shown that all three factors must be accounted together to adequately constraint the warm cloud radiative sensitivity to aerosol. LWP responds the humidity of the free atmosphere and the inversion strength (De Roode et al., 2014). It has been shown that the free atmospheric relative humidity can increase the sedimentation rate at the top of the cloud, altering the distribution of liquid throughout the cloud's vertical profile (Ackerman et al., 2004). Final results have constraints on LWP, EIS, and $RH_{700}$ to account for relationships between meteorology and LWP. For the sake of clarity, we consider the LWP separately from $RH_{700}$ and EIS, however we acknowledge that LWP is directly affected by the meteorology of the boundary layer. LWP is intrinsic to the second indirect effect, where aerosol acts to suppress precipitation and enhance the cloud lifetime, however quantifying exactly how LWP responds to aerosol has remained up for debate.

AMSR-E liquid water path, derived from the 19, 23, and 37 GHz channels, is used to separate observations into cloud state regimes (Wentz and Meissner, 2007). AMSR-E LWP is most accurate for low, marine warm clouds (Greenwald et al., 2007; Juárez et al., 2009). 99% of observations fell below a LWP of .4 $\frac{kg}{m^2}$ and analysis was restricted to observations with LWP below this limit. Since CRE is proportional to the optical depth of a cloud, which is directly related to the LWP, the sensitivity has a strong covariance with LWP (Stephens, 1978; Lee et al., 2009; Wood, 2012). Holding LWP effectively constant is therefore essential to estimating the AIE (Lohmann and Lesins, 2002). The number of LWP bins decreases from global to regional analysis due to sampling; on a global scale, seven LWP regimes are used, while on a regional scale, only four LWP regimes are used. Limits are placed to separate out the signals of low LWP clouds vs. high LWP clouds, as low clouds may be affected by evaporation-entrainment feedbacks while high LWP clouds may be affected by precipitation (Jiang et al., 2006; L'Ecuyer et al., 2009). While the environmental regimes are established on a percentile basis, cloud state regimes are set by having an increased number of bins for the lowest LWP clouds and a bin limit always set at .15 $\frac{kg}{m^2}$ to delineate clouds which are extremely unlikely to precipitate ( $< .15 \frac{kg}{m^2}$ ) and clouds more likely to precipitate ( $> .15 \frac{kg}{m^2}$ ) (L'Ecuyer et al., 2009). When environmental regimes are combined with cloud state constraints, the environmental regime limits remain constant throughout all cloud state regimes. The difference in the sensitivity of the warm cloud radiative effect to aerosol in one environmental regime versus another environmental regime at a constant LWP can therefore be more accurately attributed to aerosol.

## 2.5 Sensitivity

The warm cloud radiative sensitivity to aerosol, or $\lambda$, is defined as the linear regression of the shortwave CRE against ln(AI). While other studies have called similar metrics a susceptibility, we use the term sensitivity. The natural log of AI is used to better represent the effects of the smallest particles, which are more likely to act as CCN within a cloud (Köhler, 1936). The sensitivity is evaluated within environmental and cloud state regime frameworks on both global and regional scales. The observations are binned by 15 percentile bins of ln(AI). The AI bins are defined by the set of observations being regressed. The sensitivity is only calculated if there are 100 observations within the regime to ensure an adequate number of observations to regress against, and the linear regression Pearson correlation coefficient is greater than .4 to ensure the slope is a good fit within each regime. Throughout the study, although environmental and cloud state impacts are constrained through regimes, it cannot be stated with certainty that the observed changes in CRE are due to aerosol, only correlated with aerosol.

The unconstrained sensitivity, or the sensitivity of the warm cloud shortwave radiative effect to ln(AI) without limits on region, LWP, stability, or $RH_{700}$, is computed as:

$$\lambda_0 = -\frac{\partial CRE}{\partial \ln(AI)} \tag{4}$$

The partial derivative in this equation implies influencing factors other than aerosols should be held fixed. Here this is accomplished by evaluating the sensitivity with increasing constraints on the partial differential through regimes.

To hold the cloud state fixed, the sensitivity is found for seven distinct LWP regimes (k) and summed to yield a mean sensitivity:

$$\lambda_{LWP} = \sum_{k=1}^{N_{LWP}} \left( -\frac{\partial \text{CRE}}{\partial \ln(\text{AI})} \right)_k W_k \tag{5}$$

Where $W_k$ is fraction of observations in cloud state k:

$$W_k = \frac{\text{Number in Cloud State k}}{\text{Total Number}} \tag{6}$$

In our results, we evaluate the efficacy of increasing and decreasing the number of cloud states.

5      Similarly, the sensitivity within environmental regimes, defined by the estimated inversion strength and relative humidity of the free atmosphere, can be computed, weighted, and summed to account for meteorological covariability with ten regimes of each EIS (i) and $RH_{700}$ (j), where $W_{i,j}$ is the weighting factor for each environmental regime:

$$\lambda_{ENV} = \sum_{j=1}^{N_{RH}} \sum_{i=1}^{N_{EIS}} \left( -\frac{\partial \text{CRE}}{\partial \ln(\text{AI})} \right)_{i,j} W_{i,j} \tag{7}$$

Where $W_{i,j}$ is the fraction of observations in environmental regime i,j:

$$W_{i,j} = \frac{\text{Number in Environmental Regime i,j}}{\text{Total Number}} \tag{8}$$

By extension, both cloud and environmental conditions can be controlled via:

$$\lambda_{BOTH} = \sum_{k=1}^{N_{LWP}} \sum_{j=1}^{N_{RH}} \sum_{i=1}^{N_{EIS}} \left( -\frac{\partial \text{CRE}}{\partial \ln(\text{AI})} \right)_{i,j,k} W_{i,j,k} \tag{9}$$

Where $W_{i,j,k}$ is fraction of observations in both cloud state k and environmental regime i,j:

$$W_{i,j,k} = \frac{\text{Number in Environmental Regime i,j and Cloud State k}}{\text{Total Number}} \tag{10}$$

15      Finally, it is recognized that these bulk constraints do not fully capture all of the local factors that influence aerosol-cloud interactions. AI alone does not fully constrain the effect of aerosol composition which varies regionally. Thus, to control for these unaccounted for local effects, the sensitivity is further constrained by finding Eqn (9) on a $15°$ by $15°$ scale with four cloud state regimes (k), five regimes of stability (i), and five regimes of $RH_{700}$ (j) for each of the 152 regions (l).

$$\lambda_{ALL} = \sum_{l=1}^{N_{Reg}} \sum_{k=1}^{N_{LWP}} \sum_{j=1}^{N_{RH}} \sum_{i=1}^{N_{EIS}} \left( -\frac{\partial CRE}{\partial \ln(AI)} \right)_{i,j,k,l} w_{i,j,k,l} \tag{11}$$

Where $W_{i,j,k,l}$ is fraction of observations in region l in both cloud state k and environmental regime i,j.

## 3  Results

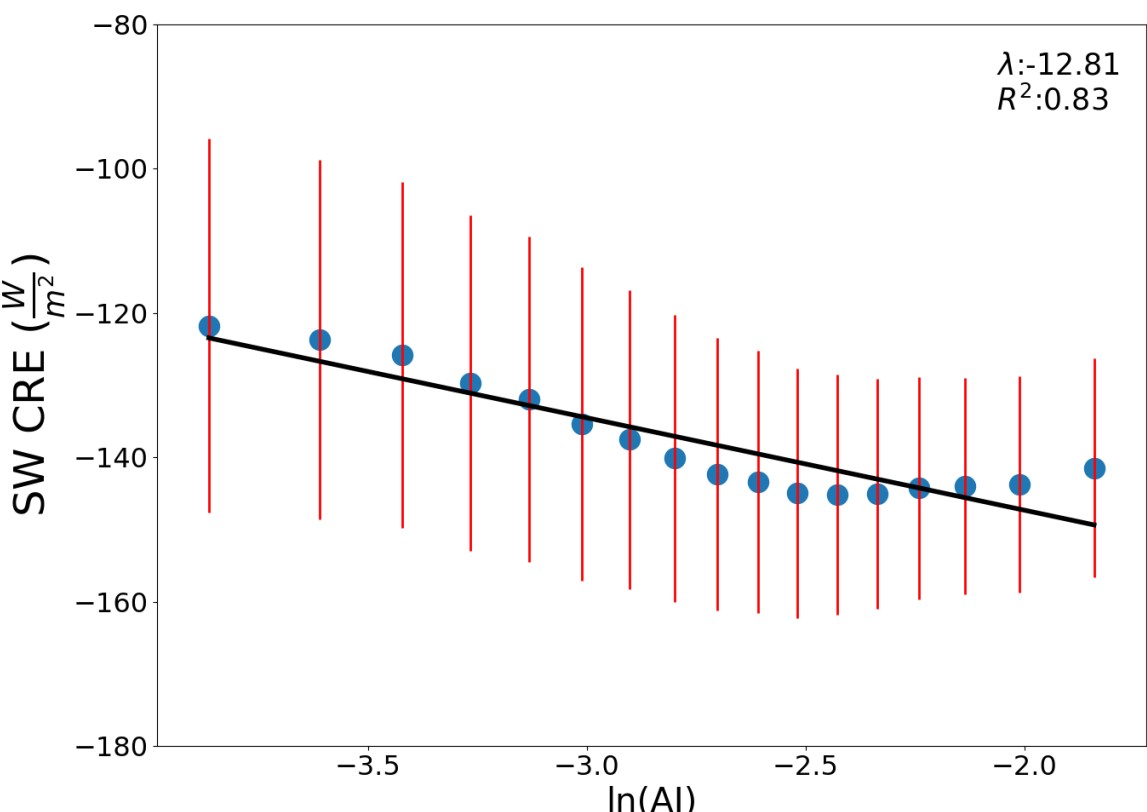

**Figure 1.** The sensitivity of CRE to aerosol ($\lambda_0$ from equation (4)) found globally from the mean SW CRE for each ln(AI) bin (blue dots) without constraints on the environment, cloud state, or region. The red lines represent the standard deviation within each bin of ln(AI).

## 3.1 Unconstrained Sensitivity

The global sensitivity of warm cloud SW forcing to aerosol without any constraints described by Equation (4) is -12.81 $\frac{W\,m^{-2}}{\ln(AI)}$ (Figure 1). This seems to capture the warm cloud AIE, after all the shortwave CRE increases with aerosol loading as expected. However, this unconstrained estimate ignores the roles of buffering and covariance. The indicated variation of SW CRE within each ln(AI) bin alludes to variation in the overall effect not captured by a single linear regression. Although the $R^2$ is high, without constraints the increase in shortwave CRE cannot be attributed to only aerosol. Furthermore, from this estimate, no information is made known on how the sensitivity varies regionally, how cloud processes affect the AIE, or whether particular cloud states may be influenced more strongly by aerosol than others.

## 3.2 Sensitivity to Cloud State

The original description of the albedo effect by Twomey (1977) specified holding the LWP of the cloud constant. Following Twomey's original hypothesis, when warm clouds are separated by LWP into cloud states, it is clear that cloud morphology plays a role in modulating the magnitude of the sensitivity (Figure 2). The total weighted, summed sensitivity is -13.12 $\frac{W\,m^{-2}}{\ln(AI)}$ for seven cloud states. From Figure 2, the lowest cloud states are less sensitive to aerosol, with a steep increase at $\sim.8\frac{kg}{m^2}$. The sensitivity increases with LWP, peaking for LWPs between .1 and .15 $\frac{kg}{m^2}$. Beyond .15 $\frac{kg}{m^2}$, the trend reverses and the sensitivity decreases with LWP, consistent with the fact that thicker clouds are already bright and less susceptible to aerosol-induced changes (Fan et al., 2016). The non-linear relationship along with the known covariance between LWP and the AIE make it a vital component of the regime framework proposed here (Feingold, 2003). Constraints on LWP limit these influences (Feingold, 2003).

The key to implementing appropriately stringent regime constraints is to determine the minimum number of cloud states required to adequately capture LWP modulation of the total sensitivity. We will be using seven cloud states throughout our global analysis as it appears to capture the impact cloud state exerts on the sensitivity while permitting ample sampling for further division of observations throughout environmental regimes. The number of cloud states are steadily increased from 3 to 7 to 11 to 23 partitions to follow a progressive increase in the number of bin limits from 4 to 8 to 12 to 24 limits, respectively. Overall, $\lambda_{LWP}$ exhibits a similar trend regardless of partitioning. The peak sensitivity for all cloud states is around .1 $\frac{kg}{m^2}$. The curve of the sensitivity and the behavior of thicker clouds is not well captured using only 3 LWP bins. The use of 7 cloud states, on the other hand, reproduces the behavior of thicker clouds and guarantees a large number of samples within each cloud state appropriate for a linear regression, especially when later partitioning by additional influences.

## 3.3 Sensitivity within Environmental Regimes

Even when separated into cloud states, aerosol impacts on warm clouds can be strongly modulated by the local environment. To account for the local meteorology, warm clouds are separated into 100 environmental regimes defined according to the local stability and free tropospheric humidity at the time they were observed (Figure 3). This approach is similar to that employed by Chen et al. (2014). Within each EIS and $RH_{700}$ regime, CERES shortwave CRE is linearly regressed against

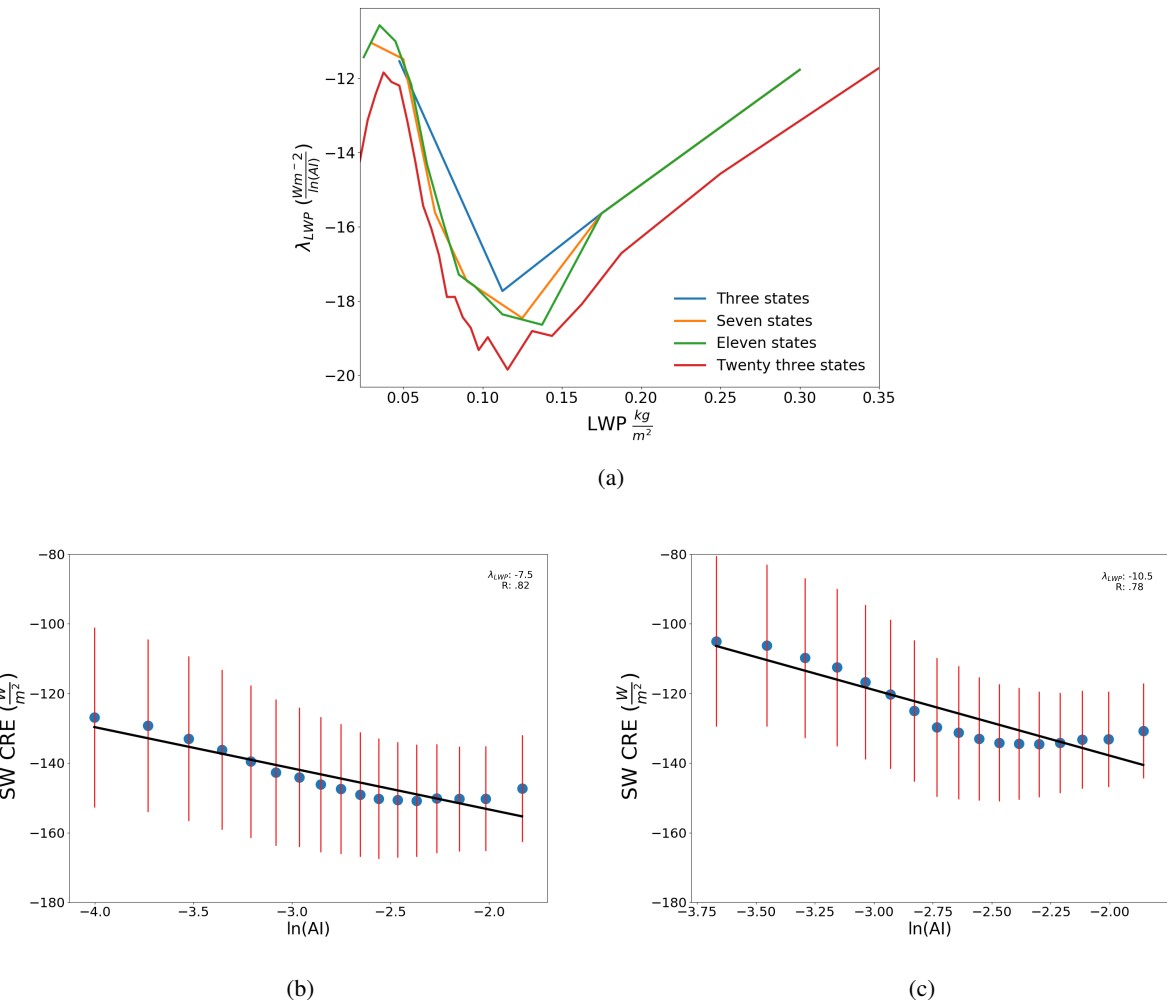

(a)

(b)

(c)

**Figure 2.** Values of the sensitivity of CRE to aerosol ($\lambda_{LWP}$ from equation (5)) for different resolutions of cloud state regimes. The weighted, summed $\lambda_{LWP}$ is -13.12 $\frac{\text{Wm}^{-2}}{\ln(\text{AI})}$ with 8 partitions. Plots of warm cloud shortwave CRE against ln(AI) are shown below for (b) thin (.04 to .06 $\frac{\text{kg}}{\text{m}^2}$) and (c) thick (.1 to .15 $\frac{\text{kg}}{\text{m}^2}$) cloud states. The red lines represent the standard deviation within each ln(AI) bin and the blue dots represent the mean SW CRE for each ln(AI) bin in plots (b) and (c).

ln(AI). The processes and resulting response are modified by the local meteorology, indicated by the change in sensitivity for different environmental regimes. Unstable environments exhibit almost no variation in sensitivity, varying by only $\sim 1 \frac{\text{W m}^{-2}}{\ln(\text{AI})}$, while stable regimes can vary by >10 ($\frac{\text{W m}^{-2}}{\ln(\text{AI})}$). The moisture content of free atmosphere influences the sensitivities in stable regimes more than unstable regimes with a clear divide at EIS = 1 K. The highest sensitivity is observed in stable regimes (EIS > 5.0K) with a moderately dry free atmosphere (Figure 3). The most sensitive warm clouds reside in environments with a moderately dry relative humidity of around 27% for an extended range of stabilities from 5 to 10 K. Warming effects (positive

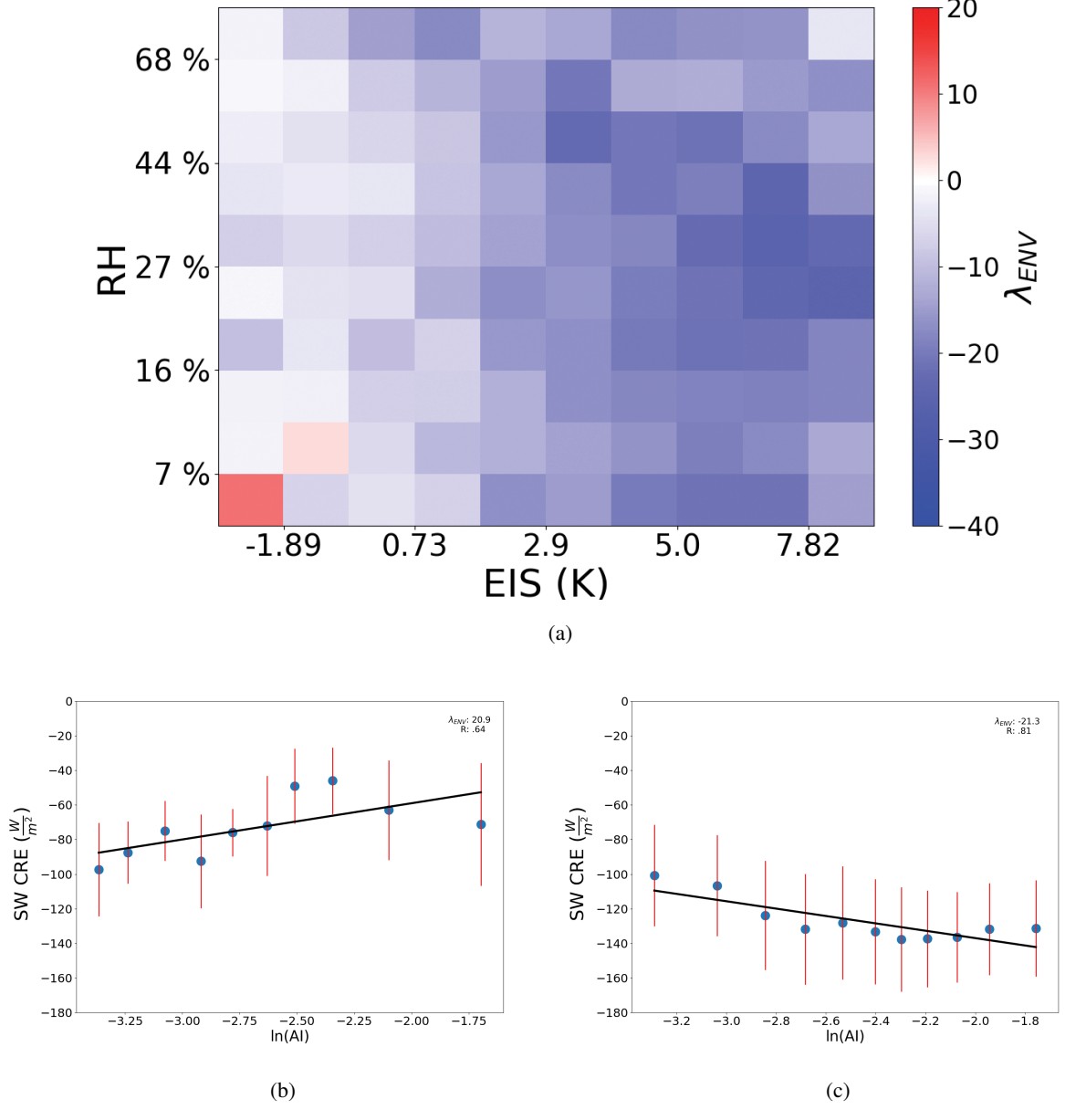

**Figure 3.** The sensitivity of CRE to aerosol ($\lambda_{ENV}$) from equation (7) evaluated with constraints on the environment. When weighted and summed following equation (7), $\lambda_{ENV}$ is -11.0 $\frac{Wm^{-2}}{\ln(AI)}$. Plots of the individual regimes from an unstable (~1K), dry environment (< 10% $RH_{700}$) (b) and stable (~6K), moist environment (>30% $RH_{700}$) (c) where the red lines represent the standard deviation of the SW CRE within each ln(AI) bin and the blue dots represent the mean SW CRE for each ln(AI) bin.

sensitives) are observed in unstable, dry environments. A warming, or reverse Twomey, effect has been noted to occur by others investigating the AIE (Chen et al., 2012, 2014). Consistent with these results, Christensen and Stephens (2011) found that up

to 1/3 of ship-tracks, occurring in primarily unstable regions, are darker than their surroundings owing to their thermodynamic feedbacks. The weighted global sensitivity calculated using Equation (7) is -11. $\frac{\text{W m}^{-2}}{\ln(\text{AI})}$ when the influence of the environment is accounted for (Figure 3).

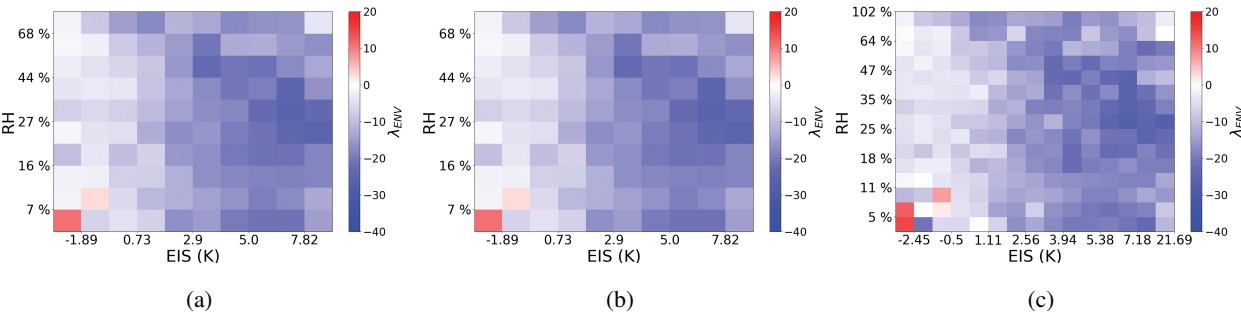

(a)           (b)           (c)

**Figure 4.** The sensitivity of the warm cloud CRE to aerosol ($\lambda_{ENV}$) found using equation 7 for environmental frameworks of a) 25 (-11.29 $\frac{\text{Wm}^{-2}}{\ln(\text{AI})}$), b) 100 (-11. $\frac{\text{Wm}^{-2}}{\ln(\text{AI})}$) and c) 225 (-10.99 $\frac{\text{Wm}^{-2}}{\ln(\text{AI})}$) regimes of EIS and RH$_{700}$.

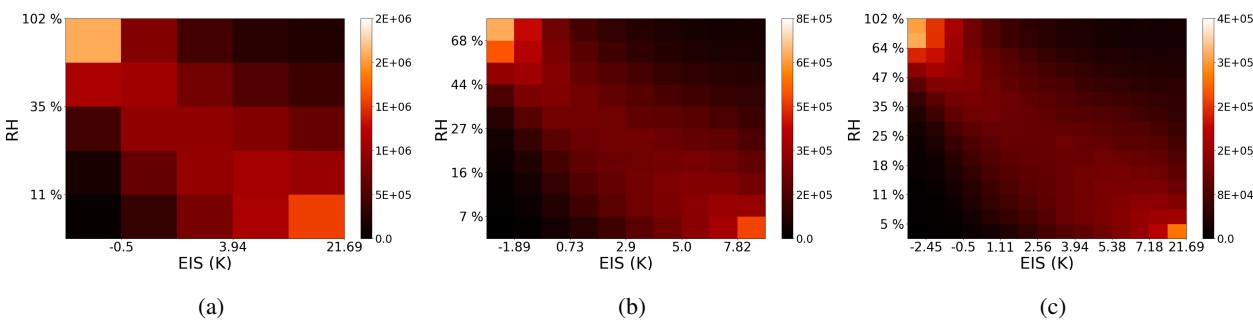

(a)           (b)           (c)

**Figure 5.** Frequency of clouds partitioned into of a) 25, b) 100, and c) 225 environmental regimes of EIS and RH$_{700}$.

The number of partitions must be narrow enough to separate the various degrees of buffering by the local meteorology and yet allow an ample number of observations per environmental regime when calculating the constrained sensitivities. To determine an optimal resolution for this dataset, the distribution of observations and sensitivity are separated into 5, 10, and 15 EIS and RH$_{700}$ partitions representing 25, 100, and 225 environmental states respectively (Figures 4, 5). The distribution of observations among environmental regimes varies smoothly with resolution (Figure 5). The minimum number of samples decreases from 35,532 to 2,707 to 757 when the resolution increases from 25 regimes to 100 regimes to 225 regimes, respectively. The mirror pattern is likely the result of the EIS in part having a slight dependence on RH$_{700}$, as the RH$_{700}$ can alter the height of the 700 mb level needed to calculate EIS. This does not impact results as this dependence is accounted for by environmental regimes. The moistest, most unstable and the driest, stablest environmental regimes always have the largest number of observations. Moist, unstable regimes are likely comprised of trade cumulus or other pre-convective cloud types in regions like the ITCZ. Dry, stable regimes are likely comprised of marine stratocumulus cloud decks off the west coast of continents.

The total sensitivity decreases as the resolution increases, from -11.29 to -11.04 to -10.99 $\frac{\text{Wm}^{-2}}{\ln(\text{AI})}$ (Figure 4). The 5 by 5 framework degrades the smoothness in $\lambda_{ENV}$ with respect to the different environmental states. The difference between the 10 by 10 and 15 by 15 estimates of sensitivity indicate that an increase in resolution after 10 partitions will lead to very little change in the overall sensitivity. However, an increased resolution decreases the number of clouds in all environmental regimes, which will be vital when the environmental regimes are further distributed among cloud states. The use of 100 regimes in analysis is appropriate to ensure proper distribution among all cloud states.

## 3.4 Accounting for Cloud and Environmental States

The preceding sections clearly demonstrate the importance of controlling for meteorological and cloud state dependencies when evaluating the sensitivity of cloud radiative effects to aerosol, however it is time to revise our framework to include both sets of constraints. Here we define three-dimensional regimes that hold LWP approximately constant while also constraining the local meteorology (Figure 6). The sensitives estimated for each of the 700 resulting regimes are shown in Figure 6. The lowest LWP cloud states show a comparatively damped maximum sensitivity than the thicker cloud states. Higher LWP clouds exhibit an increasing maximum $\lambda_{BOTH}$. The variation in magnitude between cloud states within the same environmental regimes confirms that LWP exerts a strong control in modulating the magnitude of the response and must be held constant when estimating the AIE. Mixing different cloud states in Figure 3 likely conflates differing signals, inaccurately representing the sensitivity in the most populous environmental regimes.

Again, the constrained sensitivities show distinct evidence of a darkening effect where thin clouds in the driest, most unstable environments exhibit a warming, or darkening, response to aerosol loading. Within the environmental regimes that exhibit a darkening effect, the magnitude is strongly modulated by LWP, suggesting both the expected (cooling) and opposite (warming) responses depend on LWP, $RH_{700}$, and EIS. As LWP increases, a warming $\lambda_{BOTH}$ favors increasingly moist, stable environments.

The summed and weighted sensitivity with constraints on both LWP and meteorology is -10.6 $\frac{\text{Wm}^{-2}}{\ln(\text{AI})}$. Overall, the largest sensitivity is seen in stable, moderately dry environments (Figure 6h). These environments are $\sim$ 7K of stability and $\sim$ 30% $RH_{700}$ independent of LWP. Their large sensitivity is due in part to their prominence, as most marine stratocumulus cloud decks occur in stable environments with a dry free troposphere. The weakest sensitivity occurs in unstable, dry regimes and stable, moist regimes. While these environmental conditions and cloud states are less common, discerning global warming signal with stringent constraints is significant.

These results also suggest that AIE is overestimated in approaches that do not hold the LWP approximately constant. When summed and weighted by frequency of occurrence, over almost all environmental regimes, constraining LWP damps the sensitivity (Figure 6). The difference between the LWP constrained and only environmentally constrained sensitivities reveals the strong dependence of cloud response on stability, $RH_{700}$, and LWP. In very few unstable environments, LWP constraints act to amplify the response. This effect is only observed in the the most moist and unstable or dry, stable states that have a high density of observations. LWP constrains in these regimes pulls out otherwise obstructed or buffered signals.

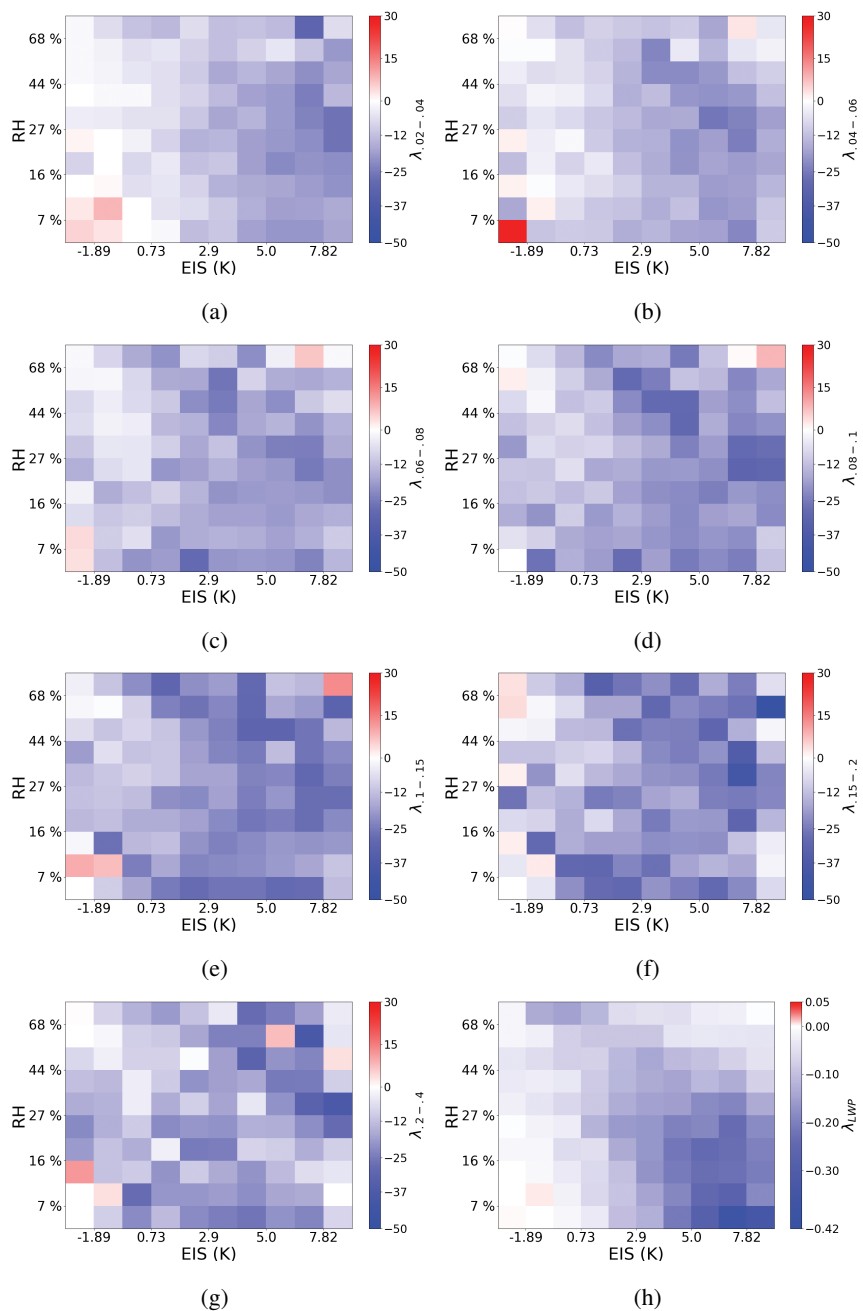

**Figure 6.** The sensitivity of CRE to aerosol ($\lambda_{BOTH}$) found with constraints on stability, $RH_{700}$ and cloud state limits of a) .02 to .04 $\frac{kg}{m^2}$ (-3.7 $\frac{Wm^{-2}}{\ln(AI)}$), b) .04 to .06 $\frac{kg}{m^2}$ (-2.2 $\frac{Wm^{-2}}{\ln(AI)}$), c) .06 to .08 $\frac{kg}{m^2}$ (-1.4 $\frac{Wm^{-2}}{\ln(AI)}$), d) .08 to .1 $\frac{kg}{m^2}$ (-1. $\frac{Wm^{-2}}{\ln(AI)}$), e) .1 to .15 $\frac{kg}{m^2}$ (-1.5 $\frac{Wm^{-2}}{\ln(AI)}$), f) .15 to .2 $\frac{kg}{m^2}$ (-.5 $\frac{Wm^{-2}}{\ln(AI)}$), and g) .2 to .4 $\frac{kg}{m^2}$ (-.4 $\frac{Wm^{-2}}{\ln(AI)}$). Panel (h) is the summed, weighted sensitivity $\lambda_{BOTH}$ within each environmental regime. The weighted, summed sensitivity is -10.6 $\frac{Wm^{-2}}{\ln(AI)}$ (sum of panel (h)). Note the colorbar for panel (h) is adjusted due to weighting.

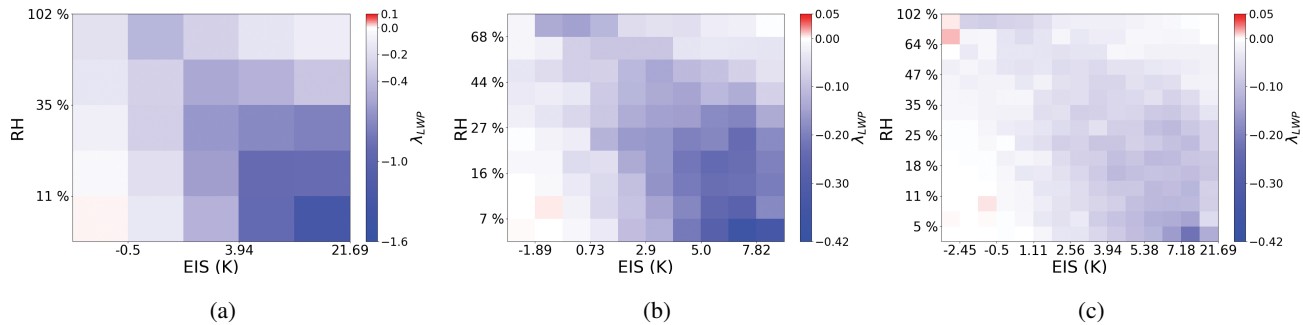

**Figure 7.** The sensitivities of CRE to aerosol from equation (9) within environmental regime resolutions of a) 5 by 5 (-10.8 $\frac{\text{Wm}^{-2}}{\ln(\text{AI})}$), b) 10 by 10 (-10.6 $\frac{\text{Wm}^{-2}}{\ln(\text{AI})}$), and c) 15 by 15 (-10.6 $\frac{\text{Wm}^{-2}}{\ln(\text{AI})}$) summed over all cloud states. Unlike all previous sensitivity estimates, these are weighted by occurrence.

To assess the effect of the resolution used to define environmental states when LWP constraints are added Figure 6h is replicated using 25, 100, and 225 environmental states (Figure 7). Sensitivity estimates are less varied (relative to Figure 3) when both the local meteorology and LWP are constrained , indicating that holding LWP fixed is essential regardless of the number of partitions of EIS and $\text{RH}_{700}$. The inclusion of LWP, however, places increasingly restrictive demands on sampling

volumes since each environmental regime must be sufficiently populated enough to allow robust sensitivities to be derived within a majority of cloud state partitions.

## 3.5    Sensitivity on Regional Scales

None of the results presented thus far have considered regional scale variability. To account for local processes and systematic differences in aerosol (e.g. composition, size, source) not captured by the bulk, global metrics above, the cloud state and

environmental regime framework is applied to 15° grid boxes from 60°S to 60°N. Regional variations in cloud sensitivity with a varying number of constraints on local meteorology and cloud state are shown in Figures 8 and 9. In the absence of constraints (Figure 8 top), the sensitivity exhibits larger variations in magnitude and sign than when cloud, environmental, or cloud and environmental constraints are in place (panels b and c and Figure 9). The unconstrained map (Figure 8 a) varies from -.53 to .77 $\frac{\text{W m}^{-2}}{\ln(\text{AI})}$ compared the most constrained map where the sensitivity of warm cloud CRE to aerosol varies only from

- .11 to .46 $\frac{\text{W m}^{-2}}{\ln(\text{AI})}$. In fact, without controlling for covarying influences of stability, entrainment, and cloud morphology, vast regions of predominantly trade cumulus clouds exhibit a darkening that reduce the globally integrated warm cloud AIE.

With constraints on only cloud state, the sensitivity shows greater variation in magnitude and sign than any other case (8 b). The tropics show an extreme darkening signal, much greater than the unconstrained case. The darkening likely occurs in the lowest, thinnest cloud state regimes and may be due to evaporation. The maximum cooling sensitivity occurs in the southern

oceans at a much larger magnitude than the unconstrained case. These signals are likely inflated since covarying meteorological factors are not fully constrained. While limiting the effects of cloud morphology on buffering and covariance is necessary, it is not sufficient for accurately resolving global AIE.

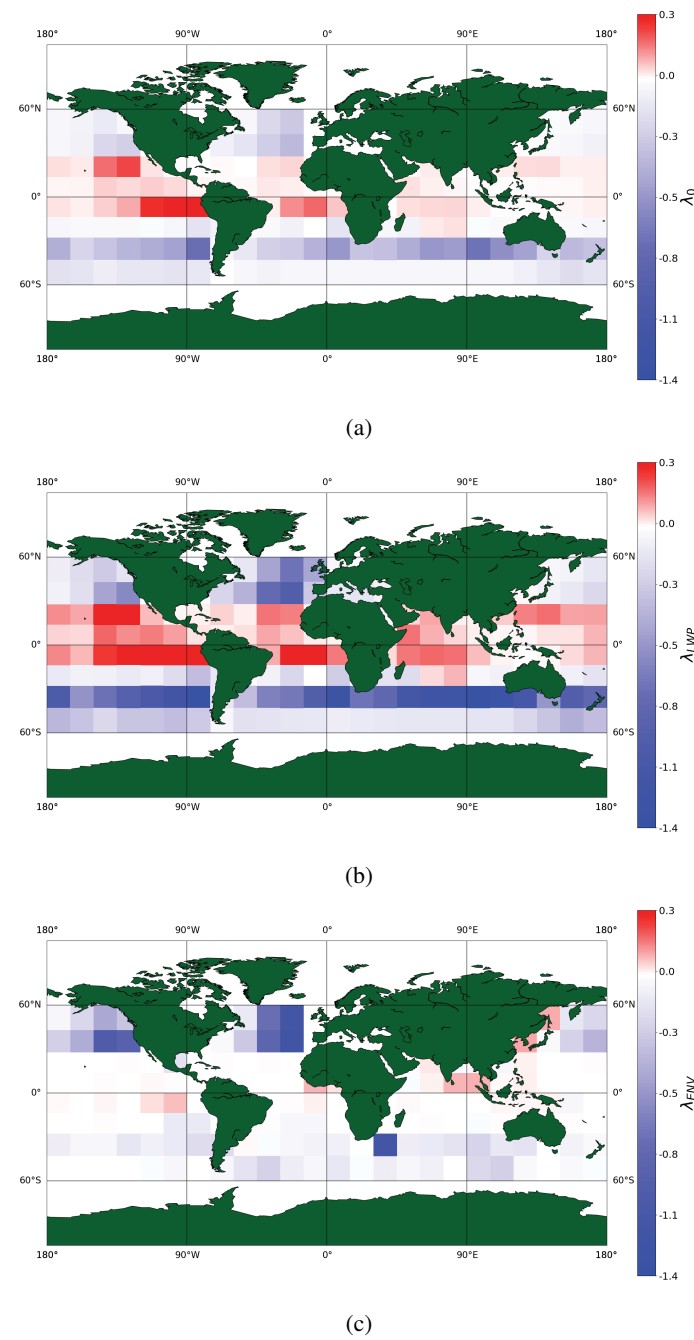

**Figure 8.** The sensitivity of CRE to aerosol evaluated regionally with (a) no regimes constraints, (b) only cloud state constraints, and (c) only environmental constraints for each $15°$ by $15°$ region. Total sensitivities are (a) -11.8, (b) -28.5, and (c) -13.8 when weighted by occurrence. $\frac{\mathrm{Wm}^{-2}}{\ln(\mathrm{AI})}$.

When constrained by local meteorological conditions alone (Figure 8 c), the sensitivity is damped in all regions. The southern ocean no longer dominates the global AIE, instead the maximum effect is seen in the north Atlantic. The warming sensitivities, or darkening, that were prevalent in the equatorial region are significantly decreased, replaced by large regions of no sensitivity. Clouds can be distributed among different LWP regimes, with differing sensitivities, that cumulatively cancel each other out even in similar environmental conditions. The environmental framework only controls for meteorological covariability, but cloud state plays a large role in modulating the sign and magnitude of effect.

The inclusion of cloud state through LWP into the regime framework is vital to adhere to the original theories of Twomey (1977) and Albrecht (1989). Both assumed the LWP to be held constant, however this cannot be true of observation based estimates of the AIE unless the LWP is explicitly limited to be approximately constant. As seen in Figure 8b, limits on LWP alone are not stringent enough to elucidate the true AIE and tend to artificially enhance sensitivities. The buffering effects of the environment and local modulating factors must also be accounted for.

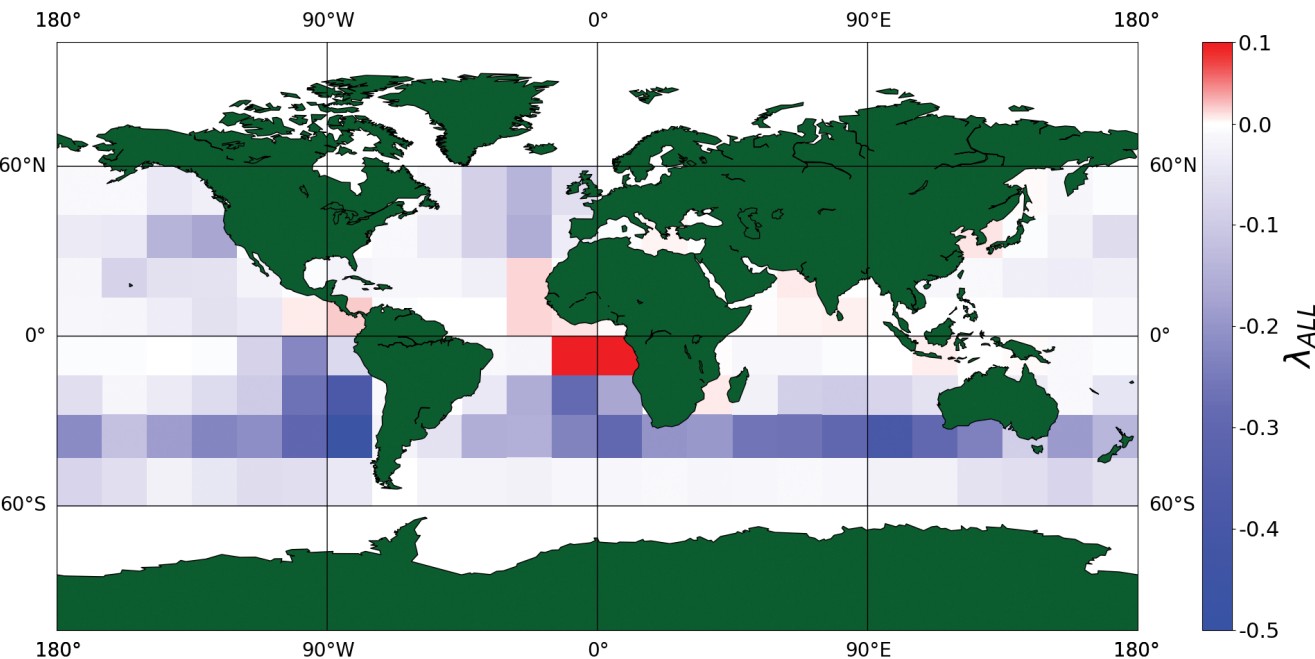

**Figure 9.** The sensitivity of CRE to aerosol ($\lambda_{ALL}$) found on a regional basis with cloud state and environmental regime constraints. The total regime weighted, global warm cloud sensitivity to aerosol perturbations is -10.13 $\frac{\text{Wm}^{-2}}{\ln(\text{AI})}$.

Including both cloud and environmental regimes limits the co-variance between aerosol, stability, cloud state, and entrainment on cloud radiative properties (Figure 9). This likely captures the true regional variation in the response of CRE to aerosol more accurately than any of the other regional estimates. The areas of strongest and weakest sensitivities exhibit coherent patterns that tends to align with distinct cloud and aerosol types. The largest sensitivities are observed in the southern subtropical oceans. Warm clouds off the coast of California exhibit a larger sensitivity with minimal constraints, i.e. with only cloud state

**Table 1.** Warm cloud shortwave radiative sensitivity to aerosol estimates with varying degrees of constraints in $\frac{\mathrm{W\,m}^{-2}}{\ln(\mathrm{AI})}$.

| | | |
|---:|:---:|:---:|
| No Constraints | $(\lambda)$ | -12.81 |
| Cloud State Constraints | $(\lambda_{LWP})$ | -13.12 |
| Environmental Constraints | $(\lambda_{ENV})$ | -11.0 |
| Cloud & Environmental Constraints | $(\lambda_{BOTH})$ | -10.6 |
| Cloud & Environmental Constraints Regionally | $(\lambda_{ALL})$ | -10.13 |

or environmental constraints. The equatorial region shows a slight warming to no effect. This is likely the region contributing to the darkening seen in the global regime framework for unstable, dry regions (Figure 6 h). The resulting global weighted mean sensitivity derived from Eqn (11) is likely representative of the complete spectrum of global shortwave warm cloud responses to aerosol.

## 4    Discussion

The sample regressions show in Figures 1, 2, and 3 illustrate the ability of constraints to reduce the variance of the observations. These constraints translate into a range of global sensitivity estimates. As constraints are applied, the sensitivity decreases from -12.81 to -10.6 to -10.13 $\frac{\mathrm{W m}^{-2}}{\ln(\mathrm{AI})}$. The decrease in total sensitivity reveals the need to constrain LWP. Holding only cloud state constant can exacerbate the signal due to mixed meteorologies, but the first order dependence of CRE on LWP requires it to be held constant. When these are applied regionally, local signals are preserved allowing the closest to truth estimate of -10.13 $\frac{\mathrm{W m}^{-2}}{\ln(\mathrm{AI})}$. This estimate is only possible through the power of sampling provided by 1.8 million satellite observations partitioned among 700 regimes, or 15,200 when further partitioned on a regional basis to represent local scale processes.

In theory, partial derivatives, such as $\frac{\partial \mathrm{CRE}}{\partial \ln(\mathrm{AI})}$, assumes other variables are held constant. The folly in treating warm clouds as only a function of aerosol is evident in Figure 8, where regionally the sensitivity of the warm cloud CRE to aerosol changes with the constraints in place, even "homogeneous" marine stratocumulus cloud deck regions. Vast areas of darkening effects are substantially moderated when the local meteorology and LWP are explicitly considered (Chen et al., 2012). These regional reversals of sensitivity to aerosols demonstrate regime-specific responses on a regional basis. LWP in particular may play a large role in determining if a cloud brightens or darkens as a result of aerosol loading.

Partitioning by regime identifies environments and cloud states that buffer, amplify, or diminish cooling. Buffering can involve any number of meteorological processes that lead to an altered response (Turner et al., 2007). For example, the local meteorology, especially $RH_{700}$, can work to inhibit or invigorate the cloud's response to aerosol (Lu and Seinfeld, 2005; Ackerman et al., 2004). Instilling limits on $RH_{700}$ should decrease any co-variance between the lifetime effect and $RH_{700}$ that could arise due to entrainment's role in cloud breakup (Kubar et al., 2015). Entrainment of drier air will force evaporation, decreasing particle size, while entrainment of moister air could have no effect or a reverse effect, increasing the number of CCN within the cloud.

Unstable regimes may act as a buffer to cloud brightening, evident when global observations are partitioned by EIS and $RH_{700}$ (Figure 6h). Unstable regimes contain pre-convective clouds (Nishant and Sherwood, 2017). Shallow cumuli, a common pre-convective cloud type found in the equatorial trade regions, are not likely to undergo the same reaction to aerosol loading as stable warm clouds like marine stratocumulus. Unstable conditions lead to strong vertical mixing and a reduced aerosol sensitivity, as activation favors strong vertical mixing in a stable environment (Cheng et al., 2017). Turbulence and vertical velocity can alter the structure of a cloud, which is especially crucial in extremely thin clouds where a redistribution of liquid water may potentially increase the likelihood of evaporation. Instability may alter the evaporation-entrainment feedback of the cloud, resulting in little to no brightening of the cloud and a severely reduced sensitivity, the result of forced evaporation reducing particle size. A reduced particle size would affect the lifetime of the cloud as well as the cloud albedo, reducing the sensitivity of the warm cloud radiative effect to aerosol loading as seen in our results for some unstable, dry regions (Jiang et al., 2006). The most unstable regimes in both Figures (4) and (6h) display the smallest sensitivities, which may be due to in-cloud turbulence decreasing the activation efficiency of the aerosol.

Without controls on the local meteorology, signals like those seen off the coast of South America, a large negative effect dominating the tropical region, may be due in part to the instability of the region and not truly reflect cloud sensitivity to aerosol loading (Figure 8). In the equatorial Atlantic off the coast of Africa, the strong decrease in CRE with respect to aerosol may not be the result of aerosol loading but that of surface winds decreasing cloud cover (Tubul et al., 2015). Surface winds were not included in analysis because the dependence of the warm cloud radiative response to aerosols depends most on LWP, $RH_{700}$, and stability, with only some regions showing a dependence on surface winds in our initial analysis. In the tropics, the warming sensitivity may be meteorologically-driven by increased frequency of trade cumuli and pre-convective clouds as stability decreases. These positive, unconstrained sensitivities are damped with environmental regime constraints (Figure 8b and 8c), however, darkening regions still appear in the fully constrained map (Figure 9), demonstrating that a substantial population of warm clouds display a true, aerosol driven darkening effect.

The role of cloud state constraints is to hold LWP approximately constant. The sensitivity to aerosol depends strongly on LWP, consistent with Wood (2012) and Ackerman et al. (2004). This relationship between LWP and aerosol-cloud-radiation interactions must be parameterized in models in order to constrain covarying effects and models must accurately simulate LWP in order to faithfully represent the cloud response (Quaas et al., 2009; Wang et al., 2011). Model parameterizations have improved the representation of warm cloud moisture fluxes, which strongly control low cloud variance, but confidence in any AIE estimates depend on cloud parameterizations continuing to improve (Guo et al., 2014).

The environmental and cloud state regimes work to limit the co-varying effects on sensitivity estimates. On both global and regional scales, the environmental constraints reveal regime-specific responses (Figures 3, 8) that allow the separation of conditions that lead to a buffered response that is especially evident in the tropical regions which undergo a sign change when meteorological constrains are in place (Figure 8) (Mülmenstädt and Feingold, 2018). In the equatorial regions, controlling for the local meteorology (Figure 8c) reduces both the sensitivity and reverse Twomey effect compared to both the unconstrained (Figure 8a) and cloud state constrained (Figure 8b) estimates. In regions that exhibit strong cloud darkening effects, a deepening boundary layer, with decreasing stability, decouple warm clouds like marine stratocumulus from the surface, fostering cloud

break up, and in turn, decreasing the cloud fraction and associated CRE of the scene. The negative sensitivities seen in the unconstrained top panel of Figure 8 are likely a result of this process, which happens simultaneously with a reduced stability, and epitomize how a single linear regression of warm cloud CRE against ln(AI) can capture meteorological effects when unconstrained (Wyant et al., 1997).

Although not explicitly controlled for, partitioning by LWP should also somewhat limit the effects of precipitation. Clouds with less than .15 $\frac{kg}{m^2}$ rarely precipitate, therefore enforcing a LWP limit at .15 $\frac{kg}{m^2}$ delineates possibly precipitating from non-precipitating clouds (L'Ecuyer et al., 2009). If precipitation does modulate aerosol-cloud interactions, the influence would only be observed in the highest LWP cloud state regimes. This is not to say precipitation is not important to aerosol-cloud interactions. In principle the regime framework presented here must be adapted to subset scenes according to the presence of
precipitation, but that is not the focus of our study.

## 5   Conclusions

Explicitly sorting satellite data by liquid water path, stability, and entrainment places increasingly stronger constraints on the partial derivative of CRE against ln(AI). This is shown to limit covariance between aerosol-cloud-radiation interactions and the environment and cloud state. In the absence of such constraints, buffering or modulation of the response by local meteorology
obfuscates estimates of the AIE (Stevens, 2007). By filtering abundant satellite observations according to the stability and relative humidity of the free atmosphere and cloud liquid water path, the local meteorology and cloud morphology are held approximately constant minimizing the chance of misinterpreting covarying of meteorology and cloud morphology as aerosol effects when regressing CRE against AI (Gryspeerdt et al., 2014). These environmental drivers are known to influence cloud extent and radiative effect, and with constraints through the use of regimes, we can better attribute changes in the CRE to
aerosol (Turner et al., 2007). Our results suggest that without constraints, the global mean AIE can be over-estimated by as much as 40% and regional variations can be artificially enhanced by as much as a factor of 2.

With environmental and cloud state constraints in place on a regional basis (Figure 9), strong, regionally specific cloud responses are identified and confidently attributed to aerosols. Clouds in the southern subtropical oceans, such as marine stratocumulus, exhibit the largest sensitivity to aerosol. Trade cumuli in the equatorial region show a much smaller, almost negligible
signal comparatively. In the northern oceans, warm cloud decks from mid-latitude cyclones through the north Atlantic interact with North American and European emissions, leading to a cooling effect.

Interestingly even after cloud state and meteorology are controlled, the analysis still reveals coherent regions of aerosol forced cloud darkening effect (Figures 6h, 9). This aggregate dimming, or reverse Twomey, effect occurs in 15% of the regions studied and appears to be a robust characteristic of low LWP clouds in unstable, dry environments. This is similar to other
observation based studies which found the same dimming effect in ~20% of warm clouds (Chen et al., 2012). Our study suggests such clouds are sufficiently abundant to consistently yield a net warming sensitivity over a substantial, coherent, region of the globe. Models must be able to recreate warm cloud responses, including the a dimming effect, if they are to accurately simulate global aerosol indirect effects.

Both on a regional and global scale, constraints reduce co-variance of sensitivity estimates (Gryspeerdt and Stier, 2012). With constraints, the sensitivity can range from .46 to -.11 $\frac{Wm^{-2}}{\ln(\text{AI})}$ on a regional scale (Figure 9), while without constraints the range increases from .77 to -.52 $\frac{Wm^{-2}}{\ln(\text{AI})}$ (Figure 8a), signaling covarying influences and buffering by the cloud distort the signal on even a regional scale. Future regime classifications should prescribe precipitation limits to further separate the effects of aerosol-cloud-precipitation interactions, which are especially important to the cloud lifetime effect, where precipitation suppression leads to a larger cloud extent and lifetime.

*Data availability.* All satellite observations and MERRA-2 reanalysis used in this study are available for download through the NASA's Data Portal at http://doi.org/10.17616/R3106C.

*Competing interests.* The authors declare that they have no conflict of interest.

## 6 Author Contributions

Alyson Douglas evaluated all data and wrote a majority of the paper. Tristan L'Ecuyer constructed the original premise, guided the evaluation, and helped with paper edits.

*Acknowledgements.* Thank you to the anonymous reviewers and Johannes Muelmenstaedt for their discussion and comments. This work was supported by CloudSat/CALIPSO Science Team grant #NNX13AQ32G. All data used in this study were obtained through the CloudSat Data Processing Center at http://www.cloudsat.cira.colostate.edu/.

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
