# Peer review of "Quantifying Variations in Shortwave Aerosol-Warm Cloud-Radiation Interactions Using Local Meteorology and Cloud State Constraints"

_Atmospheric Chemistry and Physics, 2018_

## Referee Comment (RC1) · Anonymous Referee #1 · 4 Jan 2019

General comment: The paper investigates the effect of aerosols on cloud radiative effect while taking into account the covarying influence of meteorological factors. The sensitivity of the cloud radiative effect to aerosols is derived by sorting the data by LWP, stability and entrainment. The data is retrieved from satellite observations and reanalysis, with AI serves as a proxy for the aerosols load in the atmosphere. The results show that the global aerosol indirect effect is over estimated when not accounting for the covariability. This is probably due to buffering of the clouds response by meteorology.

Major comments: 1. The authors use the term "inverse Twomey effect" which sounds

physically strange. I think that the darkening of the clouds, which refers here as "inverse Twomey effect", is the response of the LWP. The LWP decreases when cloud droplets are smaller due to evaporation (entrainment), resulting in less bright clouds. This explanation is also given in the literature that the authors cite. In addition, the "inverse Twomey effect" gets much attention in the paper, perhaps more than it should. It seems to be a rather minor effect as it occupies only a small fraction of the overall samples, as shown in most of the figures.

2. The authors write: "Constraining aerosol-cloud interactions using the local meteorology and cloud liquid water". It sounds like LWP is not part of the meteorology. However, meteorology determines boundary layer depth, and therefore also the cloud depth and LWP. Furthermore, moisture, which also controlled by meteorology in part, can alter cloud base height, and thus LWP. The authors should make it clear what they mean by meteorology.

3. The terminology used along the manuscript is inconsistent. For example, the authors use the term stability for both low level stability and inversion strength (though are similar). The same with entrainment and RH, cloud regimes and cloud states/morphologies. This is confusing.

4. Instead of using aerosols indirect effect and CRE, the authors are encourage to use the IPCC new terminology that more clearly distinguishes the key mechanisms by which anthropogenic aerosols alter the energy balance of the earth (e.g., https://doi.org/10.1175/AMSMONOGRAPHS-D-15-0033.1).

5. The captions are short and do not provide sufficient information to understand the Figures without digging into the text. Also, the captions sometimes do not present all the subplots in some of the Figures.

6. "Local Meteorology" seems to be a key factor in the study (the authors chose to have it in the title). I think that this point is not enough explained in the introduction and should be emphasized more in the conclusions.

Specific comments:

P1 L24 Provide a reference.

P2 L1-2. This is a 1.5 line paragraph. Perhaps you can discuss here the relative contribution of the cloud life time effect and cloud albedo effect.

P3 L17. I'm not sure a paper from 2014 can be considered "recent".

P3 26. Decoupling between cloud and ocean? Provide references here and in the following sentences to establish the relationship between RH and decoupling.

P3 L33. I would change effective radius to droplet size, and LWP to optical thickness.

P3 L33-35. The response of LWP to aerosols is not clear. The response is not solely due to precipitation suppression, also evaporation due to smaller drops (inverse Twomet effect?) and retrieval errors may alter observed relationships between aerosols and LWP.

Some studies show it increases, others decreases.

P4 L1. AMF?

P4 section 2.2. Please provide the spatial resolution of the data. Is the data from the different instruments is co-located to a single resolution? Why did you decide the upper threshold of LWP to be 400? Did you use optical thickness threshold to avoid additional uncertainties (see e.g. https://doi.org/10.1002/qj.2405).

If I understand correctly, the CF is determined based on a 12 km segment (P4 L29). A single open cell for example can cover 12km, which would give 100% CF, while the clear area in between cells would give 0% CF. Scaling is very important in determining the CF. You also exclude clouds with LWP<20, those are also excluded from the total CF? Such thin clouds can occupy a significant fraction over a given scene.

P5. The Equations have no numbering. Make it clear in the second eq. that the F_all

sky is only for the SW.

P5 section 2.3. You use AI as a proxy for aerosols. However, AI is retrieved only where there are no clouds. This is something that should be discussed.

P5 L22. "The cloud sensitivity" suppose to be cloud albedo sensitivity?

P6 L6 "inversion strength" is first mentioned here, which seems to be equivalent to the stability.

P6. What do the numbers above the sigma mean?

P7 section 2.6. The cloud regimes are simply LWP bins? Definition of cloud regimes is far more complex (e.g. https://doi.org/10.1002/2016JD026120).

Also note that later on in the manuscript regimes is replaced by "states" and "morphologies".

P7 L30 "Low LWP clouds are less sensitive to aerosol" - but it is the thinnest clouds that response the strongest to the Twomey effect.

P8 L9-10 Define the LWP bins.

Figure 3 (b) seems to be only a small fraction of the data.

Did you do any significant tests?

I suggest to replace the order of Figure 5 and Figure 4.

Figure 6 panel (h) is not mentioned in the caption. What does the color bar means?

P14 L1. This sentence needs context and further discussion, rather than just stating it.

P16 L5 "top" of what?

P16 L7 "stability, entrainment and cloud morphology" are equal to EIS, RH and LWP?

P16 L18. An explanation regarding the relationship between entrainment and particle

size is needed here.

Consider adding Figure 9 to Figure 8.

P7 L19. How your sensitivities (here and later) are with respect to previous studies?

P18 L16-18. I'm not sure about the context of Jiang et al. 2006 here. In their study the additional aerosols were related to enhanced evaporation, which limited the cloud life time. The study was focused on cumulus field.

P18 L19. How turbulence decreases the activation efficiency of aerosols? Turbulence can also lead to secondary nucleation due to super saturation fluctuations.

P18 L26. You mention here that wind speed can affect cloud cover. Why didn't you include also wind speed in your parameters?

P19 L19-21. I would expect decreasing stability not to decouple clouds from the surface due to more mixing. Also note that decoupling that occurs when the stability is increased can inhibit cloud breakup (https://doi.org/10.1029/2018GL078122). Please clarify. Where is the role of aerosol here?

P20 L10-11. It would be helpful to reference the relevant figures here and in the last paragraph where the sensitives are given.

---

## Referee Comment (RC2) · Anonymous Referee #2 · 7 Jan 2019

**Douglas and L'Ecuyer review**

**General comments:**
In general, I like the paper's approach to the critical problem of understanding aerosol impacts on clouds, and how they carefully dealt with co-varying meteorology. I also like how they combine measurements from different sources to get a broader picture of the system and to constrain the observations better. This paper has potential to be very important and useful for/referenced by a host of other studies using a similar methodology in the future.

That said, I had four main concerns with the paper:
1) Substantial extra work is required to clarify and explain the methodology. Another person attempting to replicate this study could probably not do so, as it is written now, and this makes it difficult for a reviewer to fully judge the value of the work. Unfortunately, by the time I reached section 3.5, the cumulative uncertainty I had with the methodology was large enough for me to have strong reservations about my ability to judge the meaningfulness of this section. That said, I do believe that if the authors answer the questions in the specific comments fully, I can better judge of the work in the next revision.
2) There is a possibility that serious errors were made regarding Fig. 5 that might have a large impact on the results. For the reasons listed in the specific comments below, I request that the authors double-check their results carefully, and if necessary, re-run the analysis.
3) A variety of confounding factors were ignored here, including the difficulties in co-locating aerosols and clouds based on AI, errors and biases in near-cloud satellite aerosol detection, and potential confounding influences of aerosol semi-direct and direct effects. These should be addressed and acknowledged.
4) Throughout the paper the conclusions tended to be a bit overstated. It should be made clearer in the manuscript that the study only focuses on a subset of data, and that this subset is not necessarily broadly representative of all conditions.

**Specific comments:**
Title, abstract, and conclusions: I suggest that the title and abstract better clarify the focus on warm marine clouds, and contain some stronger hints of the large remaining uncertainties (e.g., by adding "**Better** Understanding Aerosol-…." to the title). The reason for this suggestion is that large groups of clouds were excluded in this study, and there were some fairly major inherent uncertainties in the methodology. The study focused on daytime, single-layer, warm clouds, and as best I can tell, it only includes clouds with latitudes < 60º and over the ocean. LW forcing at night was excluded entirely. Therefore, the study cannot address several important complex cloud-radiation interactions related to aerosols. For example, the method presented here would not address ice nucleating effects or seeding effects in multi-layer clouds, which can be quite complex. The results may also not be applicable to terrestrial areas where, for example, diurnal effects of heating can be much more variable.
Introduction/Methods: Please clarify to the readers why RH and LWP, which are not independent variables, are considered separately, and essentially independently, in this study.

Methods: Please add a table where readers can quickly find what subset of clouds were included in the study. From Figures 8 and 9, it appears that terrestrial clouds and clouds poleward of 60º were eliminated. However, this is not explicitly stated in the paper. A concise central location to find which latitudes, LWPs, and temperature levels, etc. for the subset of clouds assessed in this study would be useful.

Section 2.2 or 3.5: A map of the frequency of observations of the subset of clouds compared to all clouds observed in the region would be very helpful for interpreting the relevance of this study. Are the types of clouds studied here more common in some locations than in others, and is there any geographic bias in Figures 8 and 9?

p. 4, l. 27: "*An along satellite track cloud fraction is determined by finding the average number of warm cloud pixels that satisfy these criteria (seen by CloudSat or CALIPSO, below freezing level, and LWP greater than 20 g m$^{-2}$)*" What is meant by "below freezing level"? Is that determined from MERRA2 temperature profiles below 0 ºC? Please clarify. As I interpret it, this suggests that the altitude or pressure level where the results are obtained varies by profile, and that this altitude would probably vary quite a bit over latitude and surface type. If this is the case, how do cloud altitudes in the study vary? Can we rule out that vertical variation in the clouds being studied would not add substantial error or bias the results (e.g., by introducing different aerosol types at different levels, or horizontal/vertical winds, etc.)?

p. 5, l. 3: "*The shortwave cloud radiative effect (CRE) is then defined in terms of the all sky and inferred clear sky forcings from CERES and cloud fraction from CloudSat.*" How is CloudSat cloud fraction defined? Which conditions are included in "all sky" conditions? From what I understand, situations when there are multi-layer clouds, and clouds below freezing temperatures, etc. are excluded. If this is correct, then the term "all sky" may be a little confusing, and perhaps other wording would be better.

p. 5, l. 15: Which version of MODIS is used, and why? What is the resolution of these data and how does that relate to the cloud resolution?

Section 2.3: MODIS has a variety of known issues with reliably detecting aerosols near clouds. What kind of cloud screening was used, and how sensitive are the results to this choice? It would probably be useful to note in the paper that binning the data by RH conditions could create some biases, due to aerosol swelling near high-humidity conditions typical near clouds.

Section 2.3: A major issue with using the MODIS AI, which is a column-averaged value, is its inability to co-locate aerosol layers with clouds. In some cases, for example, there may be long-range transported aerosols at high altitudes that lead to high AI values that are not representative of the aerosol conditions that the low clouds of focus in this study experience. This is of particular concern over the low-altitude marine regions of focus in this study, which are often quite clean toward the surface. Uncertainties in aerosol-cloud co-location have high potential for biasing these results in time and space, and potentially may lead to incorrect conclusions. Therefore, I recommend that the authors not only acknowledge this uncertainty here, but that they also at minimum include a short literature review of whatever is known about when and where the biases would be most likely to occur (e.g., based on CALIPSO and/or model output) for the regions of focus in this study. If the authors have the resources to do extra analysis to clarify or reduce this uncertainty as it applies to their results specifically, that would make this contribution much stronger.

Equations 2-5: On the first read-through, I was quite confused about the upper limits of summation (e.g., the number 7 in equation 2). Justifying the specific choice of those numbers might make more sense in a later section (e.g., section 2.5) than in section 2.4, where they first come up. Therefore, I suggest the authors make the equation more generalizable by having the upper limit of summation be a variable, to be assigned a value later when explanation for that value can be more logically provided. This might also help if others want to cite this method in future work, but want to use different numbers of states for their specific application. Also, please specify earlier on in the text what the upper limit of summation represents, as this was not clear in these equations and in section 2.4 in general. Moving the following text from p. 7 into section 2.4 where these limits are first introduced could help: "*The regime bounds depend on the resolution used, which is varied to establish the degree to which environmental factors must be constrained to accurately characterize sensitivity*".

Equations 2-5: As written, it is unclear how the weighting was done. Does $N_{i,j,k,l}$ imply that the data were weighted one time, or four separate times with each summation compared to $N_k$? Others may be able to get this information immediately from looking at the equation, but for readers like myself, the clarification is important.

p. 6, l.5, "*$N_k$ is the fraction of clouds that fall into LWP state k.*" Did the authors mean the "fraction of cloud profiles" instead of the "fraction of clouds"?

p. 6, l. 6: How is estimated inversion strength calculated? (note, some information is provided later, on p. 7, l. 2, but this information is not fully descriptive).

P. 7, l. 4: "*The relative humidity at 700 mb is used as a measure of the effect of entraining free tropospheric air.*" As I understand it, the RH at on vertical level is assumed to be representative of the whole vertical column up to the freezing point, or at least to provide important information for the whole column. However, RH at 700mb will be most relevant for clouds in that general altitude range. Will this bias the results, or add error? What is the variability in cloud locations? This was not quantified. Why not just use RH at the appropriate altitude ranges where the cloud layer is found?

p. 7, l. 5: "*All observations within the 5% - 95% percentiles of both EIS and RH are partitioned into regimes.*" As I understand it, one nice thing about taking the weighted mean is that you can use all of the data, and still get representative results. Thus, I don't understand why these data were excluded in the first place? (from the above statement, I believe the excluded data would equal between 10-20% of their subset of observations?) Was a similar procedure was not followed for LWP, and if so, why not?

p. 7, l. 5: "*Environmental regime limits are defined such that there are the same number observations within each percentile of either EIS or RH. The regime bounds depend on the resolution used, which is varied to establish the degree to which environmental factors must be constrained to accurately characterize sensitivity.*" I am very confused by this part of the methodology. Please define in the text what is a bound and what is a limit. I am guessing that the "regime bounds" are the same thing as the upper limits of summation in equations 2-5? Is the "regime limit" the lengths of the [i,j,k, or l] bins in equations 2-5 or something else? I also think the wording of "each percentile," which in general usage implies 1 of 100 equal groups in a dataset, may also be incorrect because it does not seem consistent with the rest of the sentence and equations 2-5. Did the authors mean bins instead of percentiles? If so, what are these bins, specifically, how were they chosen, and how does the choice of spacing affect the results? I was also confused as to

why one would want to group the same number of observations within each EIS or RH percentile [or bin?], if the results are going to be weighted later?

p. 7, l. 19: Did the authors mean "…*warm, **single-layer, marine** cloud SW*…"?  It doesn't appear that they looked at terrestrial clouds, and they stated that they excluded multi-layer cloud cases.

Fig. 1: Please describe in the Figure caption what the red lines and blue dots represent (blue dots are presumably $\lambda$, but it is best to be completely clear). In the caption, $\lambda$ is referenced. To avoid confusion, please state which $\lambda$ is being referenced (so far $\lambda_0$, $\lambda_{LWP}$, $\lambda_{ENV}$, $\lambda_{BOTH}$, and $\lambda_{ALL}$ have been defined, but no $\lambda$ without a subscript). Please also specify which $\lambda$ is being discussed in the rest of the paper as well as the symbol is used frequently. Please clarify that the $R^2$ value in Fig. 1 is describing the blue points and not the underlying distributions of all the data, because the largely overlapping red bars would suggest that in fact the correlation of the more raw data before that averaging happens is much smaller. Attaching a p-value to this and other similar figures appearing later in the paper seems appropriate.

p. 7, l. 22: "*The indicated variation of SW CRE within each ln(AI) (red bars) bin alludes to variation in the overall effect not captured by a single linear regression.*" Please describe exactly how these red bars were calculated – without knowing that, it is very hard to interpret the information the red bars are intended to convey.

Section 3.1: Many others have provided similar values to the -12.81 value provided here.  It might be good to compare this finding with findings from previous works.

Figure 2a: What cloud states are included here, and how were they derived and chosen? Some explanatory information is provided in section 3.2, but only after this Figure is referenced, which makes things confusing for the reader. It would be best if the figure could be a standalone item without requiring substantial reference to the text. The y-axis label for Fig. 2a seems to be missing, and only the units are provided. The text explaining Fig. 2a is not explicitly identified in the caption.

p. 7, l. 29: It might be helpful to reference which equation was used to derive the -13.12 value.

p. 8, l. 4: "*Constraints on LWP limit these influences.*" This is already a well-known phenomenon. The authors should probably credit here some of the other work that has previously established this finding.

p. 8, l. 6: why were 3,7,11, and 23 divisions chosen?

Section 3.2: The term "cloud state" is commonly used throughout the paper, and it is the focus on section 3.2. However, cloud state is not explicitly defined in the paper, as far as I can tell, and this is very confusing for the reader. The data in section 3.2 mostly revolve around clouds binned by LWP. Is it possible to just use LWP, instead of "cloud state"? Another minor suggestion: the authors might consider changing "cloud regimes" to "LWP bins" (if this is correct). That would be a lot easier for a casual reader of the paper to understand.

Fig. 3a: Where is the caption text describing Figure 3a? Please clarify where the -11 value in the Fig. 3 caption comes from in relationship to these figures. Is it based on the weighted mean of the data in Fig. 3a? Where is the label for the z-axis in Fig. 3a?

Figs. 3b and 3c: Please state in the caption how moist and dry environments are defined. Are these figures examples of data within individual grid cells from Fig. 3a?  If so, please state that. The red bars seem to suggest that there may be no significant differences

between any of the ln(AI) values within Fig. 3b or Fig. 3c, including at very high ln(AI) values and very low ln(AI) values?

p.10, last line: "*To account for the local meteorology, warm clouds are separated into 100 environmental regimes…*" This method seems to closely parallel the methodology of previous work (e.g., Chen et al. (2014)). It would be appropriate for the authors to reference such work here.

Section 3.3: To support the conclusions in this section, I strongly recommend that more information be provided on which colors of the grid cells of Fig. 3a are significantly different from zero. To address the signficance, the authors might consider obtaining a confidence interval around the slope for each grid cell (preferably a bootstrapped confidence interval, as that would be valid even if the data don't meet all the assumptions of a normal linear regression analysis). That would indicate whether the slope in each individual grid cell is significantly different from zero, and this information could be conveyed in the figure with markings on the grids.

Section 3.3: Since some of the cells in the figure probably have much greater sample sizes in the natural environment than others, to me, the weighted mean is probably more meaningful than the findings of individual grids, and I think it would be appropriate to stress this more in the paper.

p. 11, l. 3: "*The highest sensitivity is observed in stable regimes (EIS > 5.0) with a moderately dry free atmosphere.*" And p. 11, l. 8: "*Above 1 K, λ increases with increasing RH, while in less stable environments, RH plays only a secondary role in modulating the sensitivity.*" In Fig. 3a, I don't see evidence so far of there being higher sensitivity in drier environments, or of the latter statement at all. Were Figs. 3b,c supposed to be referenced here?  Please provide more information to substantiate these statements.

Fig. 4: In the caption, sensitivity of what?  Again, here, I think it would be really useful to note which of the grid cells are significant. Sample number in each grid cell will go down as resolution increases, and that would presumably impact the weighted mean values presented and discussed with respect to this figure, so significance would be a useful metric to help evaluate these results.

Fig. 5: This figure seems very likely to have an error. I do not see how the clouds at two extreme EIS and RH values can have the highest frequency of occurrence. If the x- and y-axis ranges were selected appropriately, one would expect the points approximately in the middle to be most frequent, and the points at the edges to be least frequent. Also, why is there such a strong mirror-like diagonal pattern in the plot?  Natural data rarely show such a distinct pattern unless the x and y variables are highly related to each other.  Please check that the data plotted here are correct.

The last 2 paragraphs of section 3.3 could probably go better in supplementary material.

Fig. 6: Again, please define in the caption which λ is being presented. Please show significance of each grid cell. Please clarify what the weighted summed sensitivity is (is it for plots a-g?). What is plotted in Fig. 6h? What meteorological constraints are plotted for each subpanel in this figure (please label that in both the caption and on the subpanels).  I am still confused about what "cloud state" even is.  Is it LWP?  Why were the ranges shown in this figure chosen?

p. 14, l. 3: "*Overall, the largest ln(AI) sensitivity is seen in stable, dry environments (Figure 6h).*" I don't see this shown in that figure.

Section 3.4, paragraphs 1-3: It seems to me that the information most relevant to the conclusions in this section is the weighted mean, and not the data shown in Figure 6. As presented, the differences between the Fig. 6 subpanels are difficult to distinguish from each other. They are also difficult to interpret, especially without information on their individual frequency of occurrence within each LWP bin and without data on their individual significances. Even if that information were presented, the figure would probably still be confusing for readers to interpret. For these reasons, the authors might want to consider changing the figure to show only the weighted means (or perhaps weighted means of the quandrant of the figures if they are trying to compare differences at high and low RH and EIS). The current plots could still be presented, e.g., in supplemental material. As an example, compare how the example Figure 1 in Zamora et al. (2018), which is analogous in many ways to the panels shown here, is simplified later in their Figure 3.

Fig. 7: Again, please double check that the frequencies of occurrence are correct.

p. 16, l. 5: "*In the absence of constraints (top), λ exhibits larger variations in magnitude and sign than when cloud, environmental, or cloud and environmental constraints are in place (panels b and c and Figure 9).*" Was Fig. 8 supposed to be referred to here? I don't see a panel b and c in Fig. 9, but these trends are not evident in Fig. 8….

Section 4: How do the values in Table 1 compare to other literature values? What other studies have looked at aerosol-meteorology co-variation and found similar results as here?

Figs. 8 and 9: Which of the pixels shown here are significantly different from zero?

p. 18, l. 1: It would be useful to also mention earlier on (e.g., methods?) that there were 1.8 million observations in the study.

**Technical comments:**

p.7, l. 6: "same number **of** observations"?

Fig. 3: Is the 11 supposed to have a period after it?

**References**

Chen, Y.-C., Christensen, M. W., Stephens, G. L. and Seinfeld, J. H.: Satellite-based estimate of global aerosol-cloud radiative forcing by marine warm clouds, Nature Geosci, 7(9), 643–646, doi:10.1038/ngeo2214, 2014.

Zamora, L. M., Kahn, R. A., Huebert, K. B., Stohl, A. and Eckhardt, S.: A satellite-based estimate of combustion aerosol cloud microphysical effects over the Arctic Ocean, Atmospheric Chemistry and Physics, 18(20), 14949–14964, doi:https://doi.org/10.5194/acp-18-14949-2018, 2018.

---

## Author Comment (AC1) · 19 Feb 2019

The paper investigates the effect of aerosols on cloud radiative effect while taking into account the covarying influence of meteorological factors. The sensitivity of the cloud radiative effect to aerosols is derived by sorting the data by LWP, stability and entrainment. The data is retrieved from satellite observations and reanalysis, with AI serves as a proxy for the aerosols load in the atmosphere. The results show that the global aerosol indirect effect is over estimated when not accounting for the covariability. This is probably due to buffering of the clouds response by meteorology.

*We thank the reviewer for taking the time to read and comment our paper. We will go through now and address each comment below.*

The authors use the term "inverse Twomey effect" which sounds physically strange. I think that the darkening of the clouds, which refers here as "inverse Twomey effect", is the response of the LWP. The LWP decreases when cloud droplets are smaller due to evaporation (entrainment), resulting in less bright clouds. This explanation is also given in the literature that the authors cite. In addition, the "inverse Twomey effect" gets much attention in the paper, perhaps more than it should. It seems to be a rather minor effect as it occupies only a small fraction of the overall samples, as shown in most of the figures.

*We originally chose the term "inverse Twomey effect" as the clouds darkening go against the common assumption of the first indirect effect, however you are correct and this may have been an poor choice of words. The microphysical pathway to the darkening is not the same as the Twomey effect. We have revised our study to show that there is a general darkening effect, but the source of the darkening, whether it be a reduced cloud fraction or reduced albedo, remains unknown.*
*To avoid any confusion over the Twomey effect and what we were calling the "inverse Twomey effect," we changed all references of "inverse Twomey effect" to darkening or warming.*

The authors write: "Constraining aerosol-cloud interactions using the local meteorology and cloud liquid water". It sounds like LWP is not part of the meteorology. However, meteorology determines boundary layer depth, and therefore also the cloud depth and LWP. Furthermore, moisture, which also controlled by meteorology in part, can alter cloud base height, and thus LWP. The authors should make it clear what they mean by meteorology.

*While boundary layer depth determines the maximum cloud depth, there are variations in the LWP of warm boundary layer clouds. Decoupling, cloud breakup, and precipitation can alter the LWP of the cloud independent of the boundary layer height. We therefore wanted to account for these processes separately from the influences of the meteorology like stability and entrainment of free atmospheric air.*

*We agree there should be more clarity on the difference between liquid water path and local meteorology. We have added "While the stability and entrainment directly affect the LWP, we consider the LWP separately from the local meteorology as it represents the cloud thermodynamics more than the local environmental conditions." in section 2.4.2 Cloud States page 7, line 9 to address the connections. We believe it is very*

*common to use the term local meteorology or meteorology and not imply liquid water path.*

**The terminology used along the manuscript is inconsistent. For example, the authors use the term stability for both low level stability and inversion strength (though are similar). The same with entrainment and RH, cloud regimes and cloud states/morphologies. This is confusing.**
*We agree that the terminology should be explained and remained more consistent. We have clarified what some statements may mean in the methodology and have stuck with a consistent terminology for each type of regime.*
*We have added "Here, EIS is calculated using MERRA-2 temperature and relative humidity profiles and indicates the stability of the boundary layer." To section 2.4.1 page 6, line 23.*
*We have added to the section 2.4.2 page 7, line 1 "Although there are other definitions of cloud regimes and cloud states used in other studies (e.g. Oreopoulos et al. (2017)), throughout ours cloud state or cloud morphology refers to the set of observations binned by liquid water path." to inform the reader of the wording we have chosen for the study.*
*We have added to section 2.4.1 page 6, line 29 "When referring to the effects of entrainment, it means the effects of RH." to inform the reader in the methods that the relative humidity reflects the effects of entrainment on the cloud.*

**Instead of using aerosols indirect effect and CRE, the authors are encourage to use the IPCC new terminology that more clearly distinguishes the key mechanisms by which anthropogenic aerosols alter the energy balance of the earth (e.g., https://doi.org/10.1175/AMSMONOGRAPHS-D-15-0033.1).**
*In our study, we are only finding the sensitivity of the clouds, not the ERFaci. We chose to focus on the methodology of distinguishing the signal of the warm cloud CRE to aerosol from other factors in this study, not to determine the radiative forcing of aerosol-cloud interactions. Our terminology is consistent with others in the field and we chose not to use IPCC terminology because we are not quantifying a forcing, only a sensitivity.*

**The captions are short and do not provide sufficient information to understand the Figures without digging into the text. Also, the captions sometimes do not present all the subplots in some of the Figures.**
*We agree our captions were too brief. We have added more detail to the captions to explain every part of the plot(s) shown.*

**"Local Meteorology" seems to be a key factor in the study (the authors chose to have it in the title). I think that this point is not enough explained in the introduction and should be emphasized more in the conclusions.**
*We agree that local meteorology should be focused on more in the introduction and have added "Constraining the local meteorology, or the characteristics of the environment around the cloud, as well as cloud type can significantly alter the*

*magnitude of the AIE compared to single, unconstrained global linear regression (Gryspeerdt et al., 2014)." to page 2, line 16.*

**Specific Comments**

**P1 L24 Provide a reference.**
*We have added (Albrecht, 1989) as a reference for that statement.*

**P2 L1-2. This is a 1.5 line paragraph. Perhaps you can discuss here the relative contribution of the cloud life time effect and cloud albedo effect.**
*These two lines are part of the first paragraph of the introduction. The ACP Discussion formatting makes it seem like it is a separate paragraph.*

**P3 L17. I'm not sure a paper from 2014 can be considered "recent"**
*We have removed recent from that sentence.*

**P3 26. Decoupling between cloud and ocean? Provide references here and in the following sentences to establish the relationship between RH and decoupling.**
*We have added "…by increasing the temperature and humidity gradients at the cloud top (Lellewen 2002)." to explain how RH affects the decoupling process.*

**P3 L33. I would change effective radius to droplet size, and LWP to optical thickness.**
*We have changed the sentence to "In his original work, Twomey postulated that cloud albedo ought to increase with aerosol provided LWP is held fixed, after 10 all, albedo is dependent on the optical depth and effective radius." replacing LWP with optical depth.*

**P4 L1. AMF?**
*We have expanded this acronym to "Amospheric Radiation Measurement Mobile Facility."*

**P4 section 2.2. Please provide the spatial resolution of the data. Is the data from the different instruments is co-located to a single resolution?**
*We have added to section 2.2 Cloud "All data is interpolated down to CloudSat's ~1km footprint." Further, in section 2.1 Data we state "The A-Train is a series of synchronized satellites which allow for collocated observations from a variety of instruments (L'Ecuyer and Jiang, 2011)."*

**Why did you decide the upper threshold of LWP to be 400? Did you use optical thickness threshold to avoid additional uncertainties (see e.g. https://doi.org/10.1002/qj.2405).**
*We chose a limit of 400 because it removes outlier cases of convective warm clouds and other thicker clouds that are not the focus of this study. Less than 5% of warm*

*clouds in our dataset had an LWP above 400. Additionally, having 400 as an upper limit reduces the impacts of warm rain on aerosol-cloud-radiation interactions.*

**If I understand correctly, the CF is determined based on a 12 km segment (P4 L29). A single open cell for example can cover 12km, which would give 100% CF, while the clear area in between cells would give 0% CF. Scaling is very important in determining the CF. You also exclude clouds with LWP**
*We have explained how we quantify cloud fraction further in section 2.2 Cloud page 5, line 9:*
*"An along-satellite track cloud fraction is determined by finding the average number of warm cloud pixels that satisfy these criteria (seen by CloudSat or CALIPSO, below the CloudSat determined freezing level, and LWP between .02 and .4 kg ) over each 12 km segment of the CloudSat track on a pixel by pixel basis, a scale that represents both the local scale length of the boundary layer and field-of-view used to define cloud radiative effects from Clouds and the Earth's Radiant Energy System (CERES) (Oke, 2002)."*
*Our cloud fraction is pixel by pixel, meaning that as cloudiness changes at a 1km scale, the cloud fraction increases or decrease by 1/12th.*

**Make it clear in the second eq. that the F_all sky is only for the SW.**
*We have changed our statement to "It is easy to show that for the shortwave radiances:" before F_all sky equation on page 5.*

**P5. The Equations have no numbering.**
*The equation numbering appears on the far right of the page. There is no way to change the formatting of this as it is set by the ACP Discussion Paper template.*

**P5 section 2.3. You use AI as a proxy for aerosols. However, AI is retrieved only where there are no clouds. This is something that should be discussed.**
*We have addressed this in section 2.3 Aerosol by adding to page 6, line 9 "While AOD and the Angstrom exponent from MODIS are not available in cloud scenes, the collocated dataset interpolates these between clear sky scenes in order to infer an AI in cloudy scenes."*

**P5 L22. "The cloud sensitivity" suppose to be cloud albedo sensitivity?**
*We have clarified this further in section 2.5 Sensitivity page 7, line 26 by adding "he warm cloud radiative sensitivity to aerosol, or λ, is defined as the linear regression of the shortwave CRE against ln(AI). While other studies have called similar metrics a susceptibility, we use the term sensitivity." This is not the cloud albedo sensitivity as ours can include effects on cloud extent/lifetime. To delineate a cloud albedo sensitivity, the indirect effect/ERFaci would have to be separated by its parts, the RFaci and cloud adjustments.*
*We have clarified throughout the study that we are deriving the warm cloud radiative sensitivity to aerosol.*

**P6 L6 "inversion strength" is first mentioned here, which seems to be equivalent to the stability.**
*You are correct. We have added "Stability of the boundary layer is indicated by the EIS." to section 2.4.1 Environmental Regimes to clarify this.*

**P6. What do the numbers above the sigma mean?**
*The numbers above sigma represent the number of regimes. I.e. we use 7 cloud state regimes, 10 regimes of EIS, and 10 regimes of RH in equation 6, while in equation 7 the number of regimes is reduced due to sampling on a regional vs. global basis to 4 cloud state regimes, 5 regimes of EIS, and 5 regimes of RH. This is common notation when using sigma (Σ) notation of summation.*

**P7 section 2.6. The cloud regimes are simply LWP bins? Definition of cloud regimes is far more complex (e.g. https://doi.org/10.1002/2016JD026120).**
*We understand that other studies have defined cloud states/regimes differently than other studies and have added to address this in section 2.4.2 page 7, line 6 "Although there are other definitions of cloud regimes and cloud states used in other studies (e.g. Oreopoulos et al. (2017)), throughout our results and analysis, cloud state or cloud state regime will refer to observations binned by liquid water path."*

**P7 L30 "Low LWP clouds are less sensitive to aerosol" - but it is the thinnest clouds that response the strongest to the Twomey effect.**
*We have rephrased our statement to reflect that based on our results, the thinnest clouds showed the lowest sensitivity. We have added to page 10, line 13 "From Figure 2, the lowest cloud states are less sensitive to aerosol, with a steep increase at ~.8 kg/m2." This is a result seen in our analysis based on observations with minimal constraints, unlike the model Twomey used which was idealized and did not include processes that could reduce the CRE of extremely thin clouds.*

**P8 L9-10 Define the LWP bins.**
*The limits of the LWP bins can be seen on the figures and would add very little if explicitly stated in the text. We have added to further clarify how we established these limits on page 7, line 15*
*"The number of LWP bins decreases from global to regional analysis due to sampling; on a global scale, seven LWP regimes are used, while on a regional scale, only four LWP regimes are used. Limits are placed to separate out the signals of low LWP clouds vs. high LWP clouds, as low clouds may be affected by evaporation-entrainment feedbacks while high LWP clouds may be affected by precipitation (Jiang et al., 2006; L'Ecuyer et al., 2009). While the environmental regimes are established on a percentile basis, cloud state regimes are set by having an increased number of bins for the lowest LWP clouds and a bin limit always set at 150 g to delineate clouds which are extremely unlikely to precipitate ( < 150 g/m2 ) and clouds more likely to precipitate ( > 150 g/m2 ) (L'Ecuyer et al., 2009)."*

**Do you do any significant tests?**

*Yes, to include the regime in analysis it must have at least 100 observations and a Pearson correlation coefficient greater than .4. These criteria are also in place when the sensitivity is found on a regional basis, where the environmental regimes are more likely to have less than 100 observations or a worse linear fit. We have added to section 2.5 Sensitivity page 7, line 31 "The sensitivity is only included if there are 100 observations within the regime and the linear regression Pearson correlation coefficient is greater than .4."*

**Figure 6 panel h is not mentioned in the caption. What does the color bar mean?**
*The colorbar for panel h is the summed, weighted sensitivity. We have added to caption for Figure 6: "Panel (h) is the summed, weighted sensitivity within each environmental regime. The weighted, summed sensitivity is -10.6 Wm−2/ln(AI) (sum of panel (h)). Note the colorbar for panel (h) is adjusted due to weighting."*

**P14 L1. This sentence needs context and further discussion, rather than just stating it.**
*We have chosen to remove this sentence.*

**P16 L5 "top" of what?**
*We have changed this to "top panel of Figure 8."*

**P16 L7 "stability, entrainment and cloud morphology" are equal to EIS, RH and LWP?**
*Yes you are correct, we use the terms stability, entrainment, and cloud morphology interchangeably with EIS, RH, and LWP respectively in the discussion. We have addressed this through earlier comments and clarified our terminology in the Methods section.*

**P16 L18. An explanation regarding the relationship between entrainment and particle size is needed here.**
*We have added to the Discussions section Page 19, line 23 "Entrainment of drier air will force evaporation, decreasing particle size, while entrainment of moister air could have no effect or a reverse effect, increasing the number of CCN within the cloud."*

**Considering adding figure 9 to figure 8.**
*We separated them to help the reader focus on figure 9, where all constraints are in place, rather than only a panel of figure 8. Figure 9 is the final focus of our discussion and therefore is better suited to be its own standalone map, rather than a panel of figure 8.*

**P18 L16-18. I'm not sure about the context of Jiang et al. 2006 here. In their study the additional aerosols were related to enhanced evaporation, which limited the cloud life time. The study was focused on cumulus field.**
*We chose to cite Jiang 2006 as it was one of the first studies to theorize an entrainment-evaporation feedback. While their findings were limited to cumulus, this*

*does not mean the process could apply to other warm clouds like the thinner cloud states of our study. We have added to the Discussions page 20, line 7 "…which would be the result of forced evaporation and reduced particle size. The reduced particle size would affect the lifetime of the cloud as well as the cloud albedo, reducing the sensitivity of the warm cloud radiative effect to aerosol loading as seen in our results for some unstable, dry regions (Jiang 2006)."*

**P18 L19. How turbulence decreases the activation efficiency of aerosols? Turbulence can also lead to secondary nucleation due to super saturation fluctuations.**
Turbulence and higher in cloud updraft speeds can increase the efficiency of aerosol activation under certain conditions. Stable boundary layers have almost a "cap" at the boundary layer top, which acts to dampen cloud growth. Unstable boundary layers are less likely to have the "cap," meaning more turbulence and higher updraft speeds lead to higher cloud tops with possibly the same amount of activation. We have added to page 20, line 4 "Unstable conditions lead to strong vertical mixing and a reduced aerosol sensitivity, as activation favors strong vertical mixing in a stable environment. Unstable local meteorologies alter the conditions of aerosol activation (Cheng 2017)." to explain the role stability plays in modulating aerosol-cloud interactions.

**P18 L26. You mention here that wind speed can affect cloud cover. Why didn't you include also wind speed in your parameters?**
*We chose to use only EIS and RH as constraints on local meteorology as they are the strongest modulators with CRE along with LWP. During initial analysis, using multivariate linear regressions, we found the highest correlations and amount of variance explained with EIS, RH, and LWP than suface wind. We have added to page 20, line 14 "Surface winds were not included in analysis because the dependence of the warm cloud radiative response to aerosols depends most on LWP, RH, and stability, with only some regions showing a dependence on surface winds in our initial analysis." to explain this reasoning.*

**P19 L19-21. I would expect decreasing stability not to decouple clouds from the surface due to more mixing. Also note that decoupling that occurs when the stability is increased can inhibit cloud breakup (https://doi.org/10.1029/2018GL078122). Please clarify. Where is the role of aerosol here?**
*The decoupling process occurs when warm marine boundary layer clouds move from a stable to less stable environment. A less stable boundary layer is more likely to have a higher boundary layer top height, increasing the chances of the cloud becoming decoupled from the surface. We have added to explain this process further to page 20, line 34 "The negative sensitivities seen in the unconstrained top panel of Figure 8 are likely a result of this process, which happens simultaneously with a reduced stability, and epitomize how a single linear regression of warm cloud CRE against ln(AI) can capture meteorological effects when unconstrained (Wyant 1997)."*

**P20 L10-11. It would be helpful to reference the relevant figures here and in the last paragraph where the sensitivities are given.**

*We have added the appropriate figure references to the Conclusions section.*

---

## Author Comment (AC2) · 19 Feb 2019

In general, I like the paper's approach to the critical problem of understanding aerosol impacts on clouds, and how they carefully dealt with co-varying meteorology. I also like how they combine measurements from different sources to get a broader picture of the system and to constrain the observations better. This paper has potential to be very important and useful for/referenced by a host of other studies using a similar methodology in the future.

*We thank the reviewer for taking the time to read and comment on our paper. We will first address the major points, and then the specific comments.*

Substantial extra work is required to clarify and explain the methodology. Another person attempting to replicate this study could probably not do so, as it is written now, and this makes it difficult for a reviewer to fully judge the value of the work. Unfortunately, by the time I reached section 3.5, the cumulative uncertainty I had with the methodology was large enough for me to have strong reservations about my ability to judge the meaningfulness of this section. That said, I do believe that if the authors answer the questions in the specific comments fully, I can better judge of the work in the next revision.

*We agree that more information should be added to help those would like to reproduce our study. The methodology section has been expanded upon. We hope that by addressing the questions outline below further rectify this issue and help the reviewer and future readers understand how they could implement a similar methodology.*

There is a possibility that serious errors were made regarding Fig. 5 that might have a large impact on the results. For the reasons listed in the specific comments below, I request that the authors double-check their results carefully, and if necessary, re-run the analysis.

*There is an inherent relationship between the estimated inversion strength (EIS) and the relative humidity of the free atmosphere (RH). To alleviate any misunderstandings of the two meteorological variables, the relationship between EIS and RH has been explained in more detail in the Methods. The EIS depends in part on the height of the 700 mb isobar, which would directly depend in part on the relative humidity of the free atmosphere (define as 700 mb). There is some covariance between these parameters that we have now tried to address. Figure 5 is correct as it simply shows that marine warm clouds exist within environmental regimes of EIS and RH. There are well known phenomenon controlling each that lead to a relationship between the two that is not the focus of this study as could be explained further in "On the relationship between stratiform low cloud cover and lower-tropospheric stability" by Wood and Bretherton 2006. All following analysis and figures are correct according to our observations and reanalysis used.*

A variety of confounding factors were ignored here, including the difficulties in co- locating aerosols and clouds based on AI, errors and biases in near-cloud satellite aerosol detection, and potential confounding influences of

**aerosol semi-direct and direct effects. These should be addressed and acknowledged.**

*These are now more distinctly addressed and acknowledged in the methods section. We agree there is some measure of uncertainty when using satellite observations to understand cloud and aerosol as clouds invariably affect near-cloud aerosol.*

**Throughout the paper the conclusions tended to be a bit overstated. It should be made clearer in the manuscript that the study only focuses on a subset of data, and that this subset is not necessarily broadly representative of all conditions.**

*The focus of the study and the conclusions drawn from the results do only apply to warm marine clouds, however these clouds are vital to understanding many different parts of the climate such as the sensitivity and radiative balance as mentioned in the introduction. A significant source of error in the IPCC's climate sensitivity is from the indirect effect. I have added more reminders in the introduction and methods that this study applies only to warm marine clouds. The importance of understanding and quantifying the warm cloud indirect effects is widely accepted. Twomey's 1977 study of the impact of pollution on Earth's albedo has been cited over 2000 times, while Albrecht's later study in 1989 has been cited over 3400 times. Aerosol impacts on continental and poleward clouds are offset by the brighter surfaces and therefore reduced impact of the indirect effect in these regions.*

**Specific Comments**

**Title, abstract, and conclusions: I suggest that the title and abstract better clarify the focus on warm marine clouds, and contain some stronger hints of the large remaining uncertainties (e.g., by adding "Better Understanding Aerosol-...." to the title). The reason for this suggestion is that large groups of clouds were excluded in this study, and there were some fairly major inherent uncertainties in the methodology. The study focused on daytime, single-layer, warm clouds, and as best I can tell, it only includes clouds with latitudes < 60° and over the ocean. LW forcing at night was excluded entirely. Therefore, the study cannot address several important complex cloud-radiation interactions related to aerosols. For example, the method presented here would not address ice nucleating effects or seeding effects in multi-layer clouds, which can be quite complex. The results may also not be applicable to terrestrial areas where, for example, diurnal effects of heating can be much more variable.**

*This is true. As stated above, warm marine cloud systems are known to exert a strong influence on climate sensitivity, but these are certainly now the only cloud type on Earth. The title has been adjusted to: "Understanding Shortwave Aerosol-Cloud-Radiation Interactions in Marine Warm Clouds Using Local Meteorology and Cloud State Constraints." We have also identified the exact clouds we are studying in the abstract.*

**Introduction/Methods: Please clarify to the readers why RH and LWP, which are not independent variables, are considered separately, and essentially independently, in this study.**

*The relative humidity of the free atmosphere (defined as 700mb) and the liquid water path from AMSR-E are independent variables. The RH is primarily a function of the vertical motion in the free atmosphere and large-scale circulations, while the LWP is primarily a function of cloud depth, stability, in-cloud microphysical processes, and other boundary layer conditions. While there may be some relationship between these quantities, both can independently modulate aerosol indirect effects … two clouds with distinct LWP may respond differently to aerosols even in similar RH environments. Thus RH does not directly control the LWP of a cloud or completely define how the SW cloud radiative effect varies with aerosol concentration.*

**Methods: Please add a table where readers can quickly find what subset of clouds were included in the study. From Figures 8 and 9, it appears that terrestrial clouds and clouds poleward of 60o were eliminated. However, this is not explicitly stated in the paper. A concise central location to find which latitudes, LWPs, and temperature levels, etc. for the subset of clouds assessed in this study would be useful.**

*We have added on page 5, line 2 "between 60◦N and 60◦S."*
*This information is provided in section 2.2 Cloud. We state in the first line of this section "…restrict analysis to single-layer, marine warm clouds between 60◦ N and 60◦ S" and "satisfy these criteria (seen by CloudSat or CALIPSO, below the CloudSat determined freezing level, and LWP between .02 and .4 kg)" when explaining the observations chosen for analysis. We feel this is too little information to warrant adding an entire table to the manuscript.*

**Section 2.2 or 3.5: A map of the frequency of observations of the subset of clouds compared to all clouds observed in the region would be very helpful for interpreting the relevance of this study. Are the types of clouds studied here more common in some locations than in others, and is there any geographic bias in Figures 8 and 9?**

*The focus on the study is to reduce the impact of influencing factors like RH, LWP, and EIS on estimating the warm cloud indirect effect. The frequency of clouds is not important, only the sensitivity of certain cloud regimes to aerosol. Including a map of cloud fraction or frequency would convey the message that the frequency is what determines the warm cloud indirect effect, when our study is focusing on how specific regimes of warm clouds independent of frequency can dominate the warm cloud radiative sensitivity to aerosol. Other studies on warm clouds note their prevalence globally.*
*We have added to the Introduction page 1, line 17 "These clouds are most prevalent off the western coasts of continents as marine stratocumulus, as trade cumulus near the tropics, and as stratus in the storm track regions (Ackerman 2018)."*

**p. 4, l. 27:** *"An along satellite track cloud fraction is determined by finding the average number of warm cloud pixels that satisfy these criteria (seen by CloudSat or CALIPSO, below freezing level, and LWP greater than 20 g m-2)"*

**What is meant by "below freezing level"? Is that determined from MERRA2 temperature profiles below 0 oC? Please clarify.**
*Freezing level is determined by the CloudSat $0^o$ isotherm from ECWMF-AUX product. Below freezing level means the entire cloud observed by CloudSat and other satellites collocated with CloudSat was contained to the layer at or below freezing level. The focus of our study is on liquid containing clouds only, not mixed phase or ice. Therefore, by limiting to clouds below freezing level, we guarantee the clouds do not contain ice or supercooled liquid.*

**As I interpret it, this suggests that the altitude or pressure level where the results are obtained varies by profile, and that this altitude would probably vary quite a bit over latitude and surface type. If this is the case, how do cloud altitudes in the study vary?**
*Clouds do vary with altitude, however by focusing on maritime liquid clouds, the variation will be limited by the boundary layer height. This varies with EIS, which we account for in our regime framework. In essence, by accounting for EIS, we are also accounting for any effects of cloud top height. Further, when the sensitivity is calculated on a regional basis, this will further constrain any small variations in height.*

**Can we rule out that vertical variation in the clouds being studied would not add substantial error or bias the results (e.g., by introducing different aerosol types at different levels, or horizontal/vertical winds, etc.)?**
*We cannot rule out that regional variation exists in aerosol type or cloud type, which is why the results are eventually found on a regional basis to account for some of this bias.*
*We have added to section 2.2 Clouds "All observations are restricted to below the freezing level of CloudSat which is determined using an ECWMF-AUX collocated reanalysis dataset and set where ECWMF determines the $0^o$ isotherm." And further on in the same paragraph we remind the readers again that observations are "below the CloudSat determined freezing level" to clarify that it is below the freezing level determined by CloudSat and not MODIS. We have also added that "Marine warm clouds fitting these parameters reside within the boundary layer." to the end of the Cloud section in the Methods to clarify these will be low-level, boundary layer clouds.*

**p. 5, l. 3: "The shortwave cloud radiative effect (CRE) is then defined in terms of the all sky and inferred clear sky forcings from CERES and cloud fraction from CloudSat." How is CloudSat cloud fraction defined? Which conditions are included in "all sky" conditions? From what I understand, situations when there are multi-layer clouds, and clouds below freezing temperatures, etc. are excluded. If this is correct, then the term "all sky" may be a little confusing, and perhaps other wording would be better.**

*Yes, as acknowledged elsewhere, this analysis is only for warm maritime clouds. The set of observations our analysis is based on is explained in detail in section 2.2 Clouds. To remind the reader that our analysis is for only a subset of clouds, we altered all "CRE" to "warm CRE" and "cloud" to "warm cloud." This is consistently mentioned further now in the results, discussion, and conclusions sections as well.*

**p. 5, l. 3: "*The shortwave cloud radiative effect (CRE) is then defined in terms of the all sky and inferred clear sky forcings from CERES and cloud fraction from CloudSat.*" How is CloudSat cloud fraction defined? Which conditions are included in "all sky" conditions?**
*Cloud fraction is defined in Section 2.2 "Cloud"*
> *"An along-satellite track cloud fraction is determined by finding the average number of warm cloud pixels that satisfy these criteria (seen by CloudSat or CALIPSO, below freezing level, and LWP greater than 20gm2) over each 12 km segment of the CloudSat track, a scale that represents both the local scale length of the boundary layer and field-of-view used to define cloud radiative effects from Clouds and the Earth's Radiant Energy System (CERES) (Oke, 2002)"*

**From what I understand, situations when there are multi-layer clouds, and clouds below freezing temperatures, etc. are excluded. If this is correct, then the term "all sky" may be a little confusing, and perhaps other wording would be better.**
*We have added "All-sky radiances from CERES are not restricted to any type of scene and include the raw radiances observed by CERES." to section 2.2.*

**p. 5, l. 15: Which version of MODIS is used, and why? What is the resolution of these data and how does that relate to the cloud resolution?**
*We have added to section 2.3 Aerosol "MODIS AI is derived from the auxiliary dataset (MOD06-1km-AUX) developed from the overlap of the CloudSat CPR footprint and the MODIS cloud mask at pixel level."*

**Section 2.3: MODIS has a variety of known issues with reliably detecting aerosols near clouds. What kind of cloud screening was used, and how sensitive are the results to this choice? It would probably be useful to note in the paper that binning the data by RH conditions could create some biases, due to aerosol swelling near high-humidity conditions typical near clouds.**
*We have added*
> *"While AOD and the Angstrom exponent from MODIS are not available in cloudy scenes, the collocated dataset interpolates these between clear sky scenes in order to infer an AI in cloudy scene. For lower cloud fraction scenes, this interpolation is more accurate, however it is possible that in higher cloud fraction scenes, the accuracy of AI is reduced. This is a source of uncertainty within our results, but with constraints on cloud state, the error of this interpolation method should be reduced. Binning by relative humidity*

when evaluating the sensitivity should reduce some bias from aerosol swelling in humid environments.
to section 2.3 Aerosol.

**Equations 2-5: On the first read-through, I was quite confused about the upper limits of summation (e.g., the number 7 in equation 2). Justifying the specific choice of those numbers might make more sense in a later section (e.g., section 2.5) than in section 2.4, where they first come up. Therefore, I suggest the authors make the equation more generalizable by having the upper limit of summation be a variable, to be assigned a value later when explanation for that value can be more logically provided. This might also help if others want to cite this method in future work, but want to use different numbers of states for their specific application. Also, please specify earlier on in the text what the upper limit of summation represents, as this was not clear in these equations and in section 2.4 in general. Moving the following text from p. 7 into section 2.4 where these limits are first introduced could help: "*The regime bounds depend on the resolution used, which is varied to establish the degree to which environmental factors must be constrained to accurately characterize sensitivity*".**
*We have added to section 2.5:*
*"Where the numbers for summation come from i.e. the number of regimes of LWP/EIS/RH."*
*"Where $N_k$ is the number of observations of cloud state k"*
*"Where $N_{i,j}$ is the number of observations within each environmental regime:"*
*"Where $N_{i,j,k}$ is the number of observations within each environmental regime when constrained further by each of the state regimes k."*
*We have also replaced the 7, 10, and 10 with LWPs, RHs, and EISs in the summation equations to clarify what bins are being summed.*
*Further, we have added to section 2.4.2 "The number of cloud states can be varied. In our results, we evaluate the efficacy of increasing and decreasing the number of cloud states."*

**p. 6, l. 6: How is estimated inversion strength calculated? (note, some information is provided later, on p. 7, l. 2, but this information is not fully descriptive).**
*Added from Wood and Bretherton 2006 the equation for EIS to section 2.4.1 Environmental Regimes.*
$$EIS = LTS - \Gamma_m^{850}(z_{700} - LCL)$$

**P. 7, l. 4: "*The relative humidity at 700 mb is used as a measure of the effect of entraining free tropospheric air.*" As I understand it, the RH at on vertical level is assumed to be representative of the whole vertical column up to the freezing point, or at least to provide important information for the whole column. However, RH at 700mb will be most relevant for clouds in that general altitude range. Will this bias the results, or add error? What is the**

variability in cloud locations? This was not quantified. Why not just use RH at the appropriate altitude ranges where the cloud layer is found?

*700 mb is the most common level used to represent the free atmosphere. Boundary layer clouds entrain free atmospheric air, so using a level like 700 mb ensures we're getting an accurate picture of the air entering the cloud layer without any contamination from the cloud layer itself in the relative humidity (Karlsson, 2010).*

**p. 7, l. 5: "*All observations within the 5% - 95% percentiles of both EIS and RH are partitioned into regimes.*" As I understand it, one nice thing about taking the weighted mean is that you can use all of the data, and still get representative results. Thus, I don't understand why these data were excluded in the first place? (from the above statement, I believe the excluded data would equal between 10-20% of their subset of observations?) Was a similar procedure was not followed for LWP, and if so, why not?**

*The tail ends of the stability and humidity spectrums were removed because we found they biased the results to the extremes. A similar approach was taken for LWP by limiting it to 20 – 400 g/m2. These results still apply for the vast majority of warm clouds.*

**p. 7, l. 5: "*Environmental regime limits are defined such that there are the same number observations within each percentile of either EIS or RH. The regime bounds depend on the resolution used, which is varied to establish the degree to which environmental factors must be constrained to accurately characterize sensitivity.*" I am very confused by this part of the methodology. Please define in the text what is a bound and what is a limit. I am guessing that the "regime bounds" are the same thing as the upper limits of summation in equations 2-5? Is the "regime limit" the lengths of the [i,j,k, or l] bins in equations 2-5 or something else? I also think the wording of "each percentile," which in general usage implies 1 of 100 equal groups in a dataset, may also be incorrect because it does not seem consistent with the rest of the sentence and equations 2-5. Did the authors mean bins instead of percentiles? If so, what are these bins, specifically, how were they chosen, and how does the choice of spacing affect the results? I was also confused as to why one would want to group the same number of observations within each EIS or RH percentile [or bin?], if the results are going to be weighted later?**

*Section 2.4.1 Environmental Regimes has been edited for clarity. We have also added "For example, with 100 environmental regimes, the observations will be binned from by 10 percentile limits of both EIS and RH. Within each row of RH within the regime framework, there are the same number observations as within each column of EIS; however within each individual regime of both EIS and RH, the number of observations is dependent on the distribution of both EIS and RH."*

**p. 7, l. 19: Did the authors mean "...*warm, single-layer, marine cloud SW...*"? It doesn't appear that they looked at terrestrial clouds, and they stated that they excluded multi-layer cloud cases.**

*We have changed it to "single-layer, marine warm cloud" in multiple places throughout the text to clarify and remind the reader the results are for a subset of clouds only.*

**Fig. 1: Please describe in the Figure caption what the red lines and blue dots represent (blue dots are presumably l, but it is best to be completely clear). In the caption, l is referenced. To avoid confusion, please state which l is being referenced (so far l0, lLWP, lENV, lBOTH, and lALL have been defined, but no l without a subscript). Please also specify which l is being discussed in the rest of the paper as well as the symbol is used frequently. Please clarify that the R2 value in Fig. 1 is describing the blue points and not the underlying distributions of all the data, because the largely overlapping red bars would suggest that in fact the correlation of the more raw data before that averaging happens is much smaller. Attaching a p-value to this and other similar figures appearing later in the paper seems appropriate.**

*All figure captions have been edited for clarity. In figure 1, we have added "with the red lines representing the standard deviation within each bin of ln(AI) and the blue dots representing the mean SW CRE for each bin." which also addresses how the red lines were calculated.*

*All lambdas have been subscripted with the correct identifier (0, LWP, ENV, BOTH, ALL).*

**Figure 2a: What cloud states are included here, and how were they derived and chosen? Some explanatory information is provided in section 3.2, but only after this Figure is referenced, which makes things confusing for the reader. It would be best if the figure could be a standalone item without requiring substantial reference to the text. The y-axis label for Fig. 2a seems to be missing, and only the units are provided. The text explaining Fig. 2a is not explicitly identified in the caption.**

*We have added to 2.4: "While the environmental regimes are established on a percentile basis, cloud state regimes are set by having an increasing number of bins for the lowest LWP clouds and a bin always set at 150 g/m2 to have a defined boundary between clouds which are extremely unlikely to precipitate ( <150 g/m2) and clouds more likely to precipitate ( >150 g/m2)."*

*We have added a better label to the y-axis of figure 2.*

*We have added more description to the caption of figure 2.*

**p. 7, l. 29: It might be helpful to reference which equation was used to derive the -13.12 value.**

*We have added where -13.12 came from.*

**p. 8, l. 4: "*Constraints on LWP limit these influences.*" This is already a well-known work that has previously established this finding.**

*You are correct. We have added a citation to work by Feingold on LWP constraints. Further citations are mentioned in the discussion as well already.*

**p. 8, l. 6: why were 3,7,11, and 23 divisions chosen?**
*We have clarified in section 3.2*
> *"We will be using seven cloud states throughout our global analysis as it appears to capture the impacts LWP has on the sensitivity while allowing ample sampling for further division of observations throughout environmental regimes. The number of cloud states are steadily increased from 3 to 7 to 11 to 23 because those follow a progressive increase in the number of bin limits from 4 to 8 to 12 to 24 limits, respectively."*

**Section 3.2: The term "cloud state" is commonly used throughout the paper, and it is the focus on section 3.2. However, cloud state is not explicitly defined in the paper, as far as I can tell, and this is very confusing for the reader. The data in section 3.2 mostly revolve around clouds binned by LWP. Is it possible to just use LWP, instead of "cloud state"? Another minor suggestion: the authors might consider changing "cloud regimes" to "LWP bins" (if this is correct). That would be a lot easier for a casual reader of the paper to understand.**
*We have changed Cloud Regimes (2.4) to Cloud States and added "Cloud states are defined as a range of liquid water paths, such that the liquid water path is held ostensibly constant."*

**Fig. 3a: Where is the caption text describing Figure 3a? Please clarify where the -11 value in the Fig. 3 caption comes from in relationship to these figures. Is it based on the weighted mean of the data in Fig. 3a? Where is the label for the z-axis in Fig. 3a?**
*Added to caption: "When weighted and summed following equation (3), $\lambda_{ENV}$ is 11.Wm−2ln(AI)."*
*Also added to end of caption: "...where the red lines represent the standard deviation of the SW CRE within each ln(AI) bin and the blue dots represent the mean SW CRE for each ln(AI) bin"*

**Figs. 3b and 3c: Please state in the caption how moist and dry environments are defined. Are these figures examples of data within individual grid cells from Fig. 3a? If so, please state that. The red bars seem to suggest that there may be no significant differences between any of the ln(AI) values within Fig. 3b or Fig. 3c, including at very high ln(AI) values and very low ln(AI) values?**
*Added to figure 3 caption: "unstable (~1K), dry environment (< 10% RH)(b) and stable (~6K), moist environment (>30% RH)"*
*The red bars are the standard deviation within each ln(AI) bin, while the blue dots are the mean warm CRE for each ln(AI) bin, as now explained in the caption. The difference between high low ln(AI) environments is focused on the mean not deviation. There is ~20 W/m2 difference in the dry, unstable case between the high and low and ~ 35 W/m2 difference in the moist, stable case. The differences are significant enough to have slopes of 10 and -25 W/m2ln(AI) for each case respectively.*

**p.10, last line:** "*To account for the local meteorology, warm clouds are separated into 100 environmental regimes...*" **This method seems to closely parallel the methodology of previous work (e.g., Chen et al. (2014)). It would be appropriate for the authors to reference such work here.**

*You are correct. We have added a citation to this work here. "This approach is similar to other approaches taken to estimate the indirect effect such as by Chen et al. 2014." Chen et al. (2014) is also currently cited in both the introduction and discussion sections.*

**Section 3.3: Since some of the cells in the figure probably have much greater sample sizes in the natural environment than others, to me, the weighted mean is probably more meaningful than the findings of individual grids, and I think it would be appropriate to stress this more in the paper.**

*We have added to section 3.3: "The results focus on contrasting individual regimes, while the discussion focuses on contrasting constraints and the weighted, summed sensitivities."*

*Our discussion section focuses on contrasting the weighted, summed values while the results focuses on how the methodology can identify regime specific responses.*

**p. 11, l. 3: "*The highest sensitivity is observed in stable regimes (EIS > 5.0) with a moderately dry free atmosphere.*" And p. 11, l. 8: "*Above 1 K, λ increases with increasing RH, while in less stable environments, RH plays only a secondary role in modulating the sensitivity.*" In Fig. 3a, I don't see evidence so far of there being higher sensitivity in drier environments, or of the latter statement at all. Were Figs. 3b,c supposed to be referenced here? Please provide more information to substantiate these statements.**

*To highlight the differences in section 3.3 we have added "The less stable regimes in figure 3 exhibit almost no variation in unstable regimes, varying by only ~1 W/m2ln(AI) while more stable regimes can vary by >10 W/m2ln(AI)."*

**Fig. 4: In the caption, sensitivity of what? Again, here, I think it would be really useful to note which of the grid cells are significant. Sample number in each grid cell will go down as resolution increases, and that would presumably impact the weighted mean values presented and discussed with respect to this figure, so significance would be a useful metric to help evaluate these results.**

*We have changed the caption beginning to "The sensitivity of the warm cloud CRE to aerosol found using equation 3 for environmental frameworks of..."*

**Fig. 5: This figure seems very likely to have an error. I do not see how the clouds at two extreme EIS and RH values can have the highest frequency of occurrence. If the x- and y-axis ranges were selected appropriately, one would expect the points approximately in the middle to be most frequent, and the points at the edges to be least frequent. Also, why is there such a strong mirror-like diagonal pattern in the plot? Natural data rarely show such a distinct pattern unless the x and y variables are highly related to each other. Please check that the data plotted here are correct.**

*We have added to section 3.3 to address the pattern:*
> *"The mirror pattern is likely the result of the EIS in part having a slight dependence on RH, as the RH can alter the height of the 700 mb level needed to calculate EIS. This does not impact results as this dependence is accounted for by environmental regimes."*

*And*
> *"The moistest, most unstable and the driest, stablest environmental regimes always have the largest number of observations. The moist, unstable regimes are likely comprised of trade cumulus or other pre-convective cloud types in unstable regions like the ITCZ. The dry, stable regimes are likely comprised of marine stratocumulus cloud decks off the coast of west coast of continents with large scale subsidence drying the free atmosphere above."*

**p. 14, l. 3: "*Overall, the largest ln(AI) sensitivity is seen in stable, dry environments (Figure 6h)*." I don't see this shown in that figure.**
*We have added to section 3.4 "These environments are ~ 7K of stability and ~ 30% RH." to pinpoint the signal.*

**p. 16, l. 5: "*In the absence of constraints (top), λ exhibits larger variations in magnitude and sign than when cloud, environmental, or cloud and environmental constraints are in place (panels b and c and Figure 9).*" Was Fig. 8 supposed to be referred to here? I don't see a panel b and c in Fig. 9, but these trends are not evident in Fig. 8....**
*We have added references to appropriate figures in 3.5*
*And also "The unconstrained map (Figure 8 a) varies from -.53 to .77 compared the most constrained map where the sensitivity of warm cloud CRE to aerosol varies only from - .11 to .46."*

**p. 18, l. 1: It would be useful to also mention earlier on (e.g., methods?) that there were 1.8 million observations in the study.**
*You are correct and we should mention this earlier. We have added to the methods "Even with these starting constraints on LWP and height, there were 1.8 million satellite observations fitting these parameters within the time period."*

**References**

Ackerman, S., Platnick, S., Bhartia, P., Duncan, B., L'Ecuyer, T., Heidinger, A., Skofronick-Jackson, G., Loeb, N., Schmit, T., and Smith, N.: Satellites see the World's Atmosphere, Meteorological Monographs, 2018.

Karlsson, J., Svensson, G., Cardoso, S., Teixeira, J., and Paradise, S.: Subtropical cloud-regime transitions: Boundary layer depth and cloud-top height evolution in models and observations, Journal of Applied Meteorology and Climatology, 49, 1845–1858, 2010.

---

## Referee Report (RR1)

**Overall comments:**
I am pleased that the authors have worked hard to address most of the comments - the manuscript is much clearer now. Many of my concerns have now been satisfactorily addressed. However, there are several key comments from the first review that were not addressed, and one comment where my suggestion was misinterpreted, for which I have added clarification. I also have a few new comments, now that I better understand the methodology.

**Comments not addressed from the first review:**

"Section 2.3: A major issue with using the MODIS AI, which is a column-averaged value, is its inability to co-locate aerosol layers with clouds. In some cases, for example, there may be long-range transported aerosols at high altitudes that lead to high AI values that are not representative of the aerosol conditions that the low clouds of focus in this study experience. This is of particular concern over the low-altitude marine regions of focus in this study, which are often quite clean toward the surface. Uncertainties in aerosol-cloud co-location have high potential for biasing these results in time and space, and potentially may lead to incorrect conclusions. Therefore, I recommend that the authors not only acknowledge this uncertainty here, but that they also at minimum include a short literature review of whatever is known about when and where the biases would be most likely to occur (e.g., based on CALIPSO and/or model output) for the regions of focus in this study. If the authors have the resources to do extra analysis to clarify or reduce this uncertainty as it applies to their results specifically, that would make this contribution much stronger."

"Equations 2-5: As written, it is unclear how the weighting was done. Does $N_{i,j,k,l}$ imply that the data were weighted one time, or four separate times with each summation compared to $N_k$? Others may be able to get this information immediately from looking at the equation, but for readers like myself, the clarification is important."

"Section 3.3: To support the conclusions in this section, I strongly recommend that more information be provided on which colors of the grid cells of Fig. 3a are significantly different from zero. To address the significance, the authors might consider obtaining a confidence interval around the slope for each grid cell (preferably a bootstrapped confidence interval, as that would be valid even if the data don't meet all the assumptions of a normal linear regression analysis). That would indicate whether the slope in each individual grid cell is significantly different from zero, and this information could be conveyed in the figure with markings on the grids."

"Fig. 4: Again, here, I think it would be really useful to note which of the grid cells are significant. Sample number in each grid cell will go down as resolution increases, and that would presumably impact the weighted mean values presented and discussed with respect to this figure, so significance would be a useful metric to help evaluate these results."

"Fig. 1: Please clarify that the $R^2$ value in Fig. 1 is describing the blue points and not the underlying distributions of all the data, because the largely overlapping red bars would suggest that in fact the correlation of the more raw data before that averaging happens is much smaller.

Attaching a p-value to this and other similar figures appearing later in the paper seems appropriate."

"Section 3.1: Many others have provided similar values to the -12.81 value provided here. It might be good to compare this finding with findings from previous works."

p.7, l. 6: "same number **of** observations"?

Fig. 3: Is the 11 supposed to have a period after it?

**Comments on the responses:**

**Introduction/Methods: Please clarify to the readers why RH and LWP, which are not independent variables, are considered separately, and essentially independently, in this study.**

> *The relative humidity of the free atmosphere (defined as 700mb) and the liquid water path from AMSR-E are independent variables. The RH is primarily a function of the vertical motion in the free atmosphere and large-scale circulations, while the LWP is primarily a function of cloud depth, stability, in-cloud microphysical processes, and other boundary layer conditions. While there may be some relationship between these quantities, both can independently modulate aerosol indirect effects ... two clouds with distinct LWP may respond differently to aerosols even in similar RH environments. Thus RH does not directly control the LWP of a cloud or completely define how the SW cloud radiative effect varies with aerosol concentration.*

> Thanks for the explanation, I think I now better understand the study's methodology. The term "RH" is used throughout the paper, and at various points I thought that the authors meant the different RH values in the vertical column, rather than RH at the 700 mb pressure level. To avoid others being similarly confused, it would helpful to more clearly differentiate between the two in the text. The authors might consider using a term such as $RH_{700}$ instead of RH, for example, as appropriate (e.g., in the figures, and in equations, as well as in the text). I completely missed this differentiation in the methods, for example, when it is written that, "*When referring to the effects of entrainment, it means the effects of RH. All observations within the 5% - 95% percentiles of both EIS and RH are partitioned into regimes of percentile limits.*" I also missed this in the introduction, for example as written here: "*Including RH in aerosol sensitivity studies accounts for some decoupling influence. Models affirm the effects of entrainment on the cloud layer depend in part on RH, as LES have shown RH moderates cloud feedbacks in low warm cloud simulations (Van der Dussen et al., 2015).*" Others might miss that too.

> It might be worth clarifying that the Van der Dussen paper referenced above focused on the **difference** in RH at the surface and 700 mb, and not specifically on RH at 700 mb alone, as was done in this study.

So now, taking a step back, what the authors are saying above is that entrainment (if RH at 700 mb really is a good proxy for entrainment, see comment below) is mostly independent from LWP. It is still not obvious to me that LWP and entrainment should be independent - if you have dry air entering the marine boundary layer, wouldn't that tend to reduce LWP on average? Maybe the authors should just plot the LWP vs. RH at 700 mb variables for the dataset to demonstrate the relationship (or lack thereof).

**P. 7, l. 4: "*The relative humidity at 700 mb is used as a measure of the effect of entraining free tropospheric air.*" As I understand it, the RH at on vertical level is assumed to be representative of the whole vertical column up to the freezing point, or at least to provide important information for the whole column. However, RH at 700mb will be most relevant for clouds in that general altitude range. Will this bias the results, or add error? What is the variability in cloud locations? This was not quantified. Why not just use RH at the appropriate altitude ranges where the cloud layer is found?**

> *700 mb is the most common level used to represent the free atmosphere. Boundary layer clouds entrain free atmospheric air, so using a level like 700 mb ensures we're getting an accurate picture of the air entering the cloud layer without any contamination from the cloud layer itself in the relative humidity (Karlsson, 2010).*

It helps that I now understand the methodology a bit better. However, I am still a little confused. I can see how RH at 700 mb tells one something about how dry the free tropospheric air is that hypothetically could be entrained, and I know there are studies that show that RH at 700 mb is in fact correlated with various cloud properties. However, I am not sure that means that RH at 700 mb is a close proxy for actual entrainment. The authors might consider restating the description of what meteorological factor(s) that RH at 700 mb represents in a more specific way throughout the paper.

**Fig. 5: This figure seems very likely to have an error. I do not see how the clouds at two extreme EIS and RH values can have the highest frequency of occurrence. If the x- and y-axis ranges were selected appropriately, one would expect the points approximately in the middle to be most frequent, and the points at the edges to be least frequent. Also, why is there such a strong mirror-like diagonal pattern in the plot? Natural data rarely show such a distinct pattern unless the x and y variables are highly related to each other. Please check that the data plotted here are correct.**

> *There is an inherent relationship between the estimated inversion strength (EIS) and the relative humidity of the free atmosphere (RH). To alleviate any misunderstandings of the two meteorological variables, the relationship between EIS and RH has been explained in more detail in the Methods. The EIS depends in part on the height of the 700 mb isobar, which would directly depend in part on the relative humidity of the free atmosphere (define as 700 mb). There is some covariance between these parameters that we have now tried to address. Figure 5 is correct as it simply shows that marine warm clouds exist within environmental regimes of EIS and RH. There are well known phenomenon controlling each that lead to a relationship between the two that is not the*

*focus of this study as could be explained further in "On the relationship between stratiform low cloud cover and lower-tropospheric stability" by Wood and Bretherton 2006. All following analysis and figures are correct according to our observations and reanalysis used.*

*We have added to section 3.3 to address the pattern: "The mirror pattern is likely the result of the EIS in part having a slight dependence on RH, as the RH can alter the height of the 700 mb level needed to calculate EIS. This does not impact results as this dependence is accounted for by environmental regimes."*

*And "The moistest, most unstable and the driest, stablest environmental regimes always have the largest number of observations. The moist, unstable regimes are likely comprised of trade cumulus or other pre-convective cloud types in unstable regions like the ITCZ. The dry, stable regimes are likely comprised of marine stratocumulus cloud decks off the coast of west coast of continents with large scale subsidence drying the free atmosphere above."*

Ok, thanks, the new text is very helpful in better understanding this figure. But the response above has led to some new confusion on equations 6-8. If RH at 700 mb is used to define EIS, one of the variables being integrated is included in another variable being integrated separately in the same equation. Does this method then make sense, and should we be worried about co-dependency resulting in spurious results? Maybe it matters, maybe not, or maybe it does matter a bit, but logistically this is the best that can be done with the data available. Either way, perhaps this issue should be addressed in the text.

**Section 2.2 or 3.5: A map of the frequency of observations of the subset of clouds compared to all clouds observed in the region would be very helpful for interpreting the relevance of this study. Are the types of clouds studied here more common in some locations than in others, and is there any geographic bias in Figures 8 and 9?**

*The focus on the study is to reduce the impact of influencing factors like RH, LWP, and EIS on estimating the warm cloud indirect effect. The frequency of clouds is not important, only the sensitivity of certain cloud regimes to aerosol. Including a map of cloud fraction or frequency would convey the message that the frequency is what determines the warm cloud indirect effect, when our study is focusing on how specific regimes of warm clouds independent of frequency can dominate the warm cloud radiative sensitivity to aerosol. Other studies on warm clouds note their prevalence globally.*

*We have added to the Introduction page 1, line 17 "These clouds are most prevalent off the western coasts of continents as marine stratocumulus, as trade cumulus near the tropics, and as stratus in the storm track regions (Ackerman 2018)."*

My apologies if I was not clear enough here. The suggestion was not for the authors to include a CF or a total cloud frequency map. Instead, the suggestion was that they

include a map that shows how often their subset of clouds occurs relative to the total cloud observation number.

Please keep in mind that a lot of data were excluded in this study. After excluding all clouds over the ocean at temperatures below 0 °C, and all multi-layer clouds, another ~20% or so of the remaining data was excluded by removing the remaining most extreme 10% of the each of the EIS and RH at 700 mb data, and any LWP data below 20 g/m$^2$ or above 400 g/m$^2$.

Currently, the reader does not have much context for where these data were excluded, but it would not be unreasonable to guess that the excluded data are not randomly distributed throughout the sampling region. Just as an extreme hypothetical example, what if the RH criteria removed 5% of the data in one grid box, but 80% of the data in another? The above suggested map would provide the community with important context for how relevant Figure 9 (the aerosol sensitivity range estimate for this subset of clouds) is at a given region of the planet.

**Comments on the new text:**
eq. 3: It would be helpful to define LCL and $z_{700}$.

p. 6, l. 13: "*Binning by relative humidity when evaluating the sensitivity should reduce some bias from aerosol swelling in humid environments.*"

> Wouldn't that actually add to the bias, because swelling occurs more in high RH environments, so those grids would have a consistently higher apparent AI relative to the other grids?

---

## Author Response (AR2)

Reply to Report 1

**Section 2.3: A major issue with using the MODIS AI, which is a column-averaged value, is its inability to co-locate aerosol layers with clouds. In some cases, for example, there may be long- range transported aerosols at high altitudes that lead to high AI values that are not representative of the aerosol conditions that the low clouds of focus in this study experience. This is of particular concern over the low-altitude marine regions of focus in this study, which are often quite clean toward the surface. Uncertainties in aerosol-cloud co-location have high potential for biasing these results in time and space, and potentially may lead to incorrect conclusions. Therefore, I recommend that the authors not only acknowledge this uncertainty here, but that they also at minimum include a short literature review of whatever is known about when and where the biases would be most likely to occur (e.g., based on CALIPSO and/or model output) for the regions of focus in this study. If the authors have the resources to do extra analysis to clarify or reduce this uncertainty as it applies to their results specifically, that would make this contribution much stronger.**

> *We understand the hesitancy about using aa column averaged aerosol concentration proxy. These are valid concerns that column measures like AI may not always be representative of the environment near cloud but, owing to the challenges of making global aerosol measurements, AI is very commonly used in observational studies of aerosol indirect effects and there are a number of reasons to think this metric may provide useful results. We have added to section 2.3 Aerosol:*
>
> > *AI can be affected by aerosol swelling in the most humid environments. All results have some amount of uncertainty due to this effect (Lobe and Schuster, 2008). This is minimized in the driest RH regimes, however the most humid RH regimes may be affected by aerosol swelling. The effect will be largest in the cloudiest regions such as the marine stratocumulus decks in the South Atlantic, Southeast Pacific, and off the California coast because aerosol measurements near clouds (~15 km) are subjected to the largest amount of swelling (Christensen et al. 2017). Another source of uncertainty is that the aerosol may not be located at the same height as the warm, boundary layer clouds we are evaluating. Aerosol should ideally be located near the cloud base in order to be fully activated and initiate the indirect effect (Chen et al. 2018). It has been suggested using AI underestimates the strength of the indirect effect; our estimates of sensitivity of the warm cloud radiative effect to aerosol could be thought of a lower bound on the warm cloud indirect effect sensitivity (Penner et al. 2011).*

**Equations 2-5: As written, it is unclear how the weighting was done. Does $N_{i,j,k,l}$ imply that the data were weighted one time, or four separate times with each summation compared to $N_k$? Others may be able to get this**

**information immediately from looking at the equation, but for readers like myself, the clarification is important.**

> *Sorry for the confusion, this was an error on our part. I've updated the Methods section 2.5 Sensitivity to clarify the weighting methods replacing N with W (for weight).*
> *Now sequentially, 2.5 reads:*
>> *"Where Wk is fraction of observations in cloud state k."*
>> *"Where Wi,j is the fraction of observations in environmental regime i,j:"*
>> *"Where Wi,j,k is fraction of observations in both cloud state k and environmental regime i,j:"*
>> *"Where Wi,j,k,l is fraction of observations in region l in both cloud state k and environmental regime i,j."*
> *The weights change as constraints are added and/or the number of regimes is modified (e.g. 25 vs. 100 vs. 225).*

**Fig. 1: Please clarify that the R2 value in Fig. 1 is describing the blue points and not the underlying distributions of all the data, because the largely overlapping red bars would suggest that in fact the correlation of the more raw data before that averaging happens is much smaller.**
**Attaching a p-value to this and other similar figures appearing later in the paper seems appropriate.**

> *We have updated the caption of Figure 1 to say "The sensitivity of CRE to aerosol ($\lambda_0$ from equation (4)) found globally from the mean SW CRE for each ln(AI) bin (blue dots) without constraints on the environment, cloud state, or region. The red lines represent the standard deviation within each bin of ln(AI)."*
> *We have also added the slope and R values for each of the example scatterplots in Figures 2 and 3. We have chosen not explicitly add the p value, but the p values for Figures 2 and 3 subplots are:*
>> *Figure 2 b: 1.04 e-05*
>> *Figure 2 c: 4.9 e-05*
>> *Figure 3 b: .04*
>> *Figure 3 c: .002*

**Section 3.1: Many others have provided similar values to the -12.81 value provided here. It might be good to compare this finding with findings from previous works.**

> *Our study uses ln(AI), which is not often used outside of satellite-based aerosol-cloud interaction studies. Further, our $\lambda$ is kept in units per ln(AI) and is not a measure of the forcing. Many studies skip showing a sensitivity and only display the total forcing either regionally or globally (ERFaci). Comparing our sensitivity to other studies which use $r_e$ or albedo instead of the warm cloud CRE as we do is not possible. Thus we have not compared our sensitivity to others because it is not directly comparable unless we explicitly seek out the*

*unpublished sensitivity estimates from other satellite-based studies which use both the warm cloud CRE and ln(AI).*

**Response to Comments on the Reponses**

*We understand that the reader could forget that it is only RH at 700 mb and have replaced RH with $RH_{700}$.*

*We have changed the reference to Van der Dussen's paper to explain it was the difference in relative humidity in the Introduction.*

*EIS doesn't directly use the RH at 700 mb, it uses the height of the 700 mb level. The height of the 700 mb level can be influenced by some of the same processes that affect the relative humidity at this level, but is not directly the same as $RH_{700}$. We have added to section 2.4.1 Environmental Regimes*

> *Some processes involved in altering the height at 700 mb will also affect $RH_{700}$ and vice versa, therefore there is some covariability between our two meteorological variables.*

*We have created a map that shows where the extremes in observations have been removed (top/bottom 10% of EIS, RH and above 400 g/m2 of LWP). The map below is the percentage of observations in each region that met our selection criteria compared to the overall number of observations of the region. The darker the blue, the less number of observations that met our criteria. As you can see, the largest differences are in a select few coastal regions. Our focus is on warm clouds, not continental, orographic, or mid-latitude cyclones, therefore it seems that the constraints in place may have removed some cases of these clouds from our dataset. We stand by our set of constraints especially after seeing where observations have been removed.*

[Figure]

**Reply to Report 2**

**1. LWP vs meteorology. I am still not convinced why LWP is not meteorology. The authors write in their response to reviewer 1 that "While boundary layer depth determines the maximum cloud depth, there are variations in the LWP of warm boundary layer clouds. Decoupling, cloud breakup, and precipitation can alter the LWP of the cloud independent of the boundary layer height. We therefore wanted to account for these processes separately from the influences of the meteorology like stability and entrainment of free atmospheric air." The mentioned processes can be also meteorology related: decoupling can occur due to entrainment and warm advection, cloud breakup can be related to enhanced entrainment and drying, mixing due to wind shear, etc. and precipitation was shown to related to cloud depth.**
**Furthermore, in section 2.4.2 page 7, line 9 the authors write "we consider the LWP separately from the local meteorology as it represents the cloud thermodynamics more than the local environmental conditions" - thermodynamics are derived from environmental conditions. Perhaps the term mircophysical processes fits better what the authors mean.**
**The LWP can indeed be changed due to aerosols, however the sing of the response varies between studies (e.g.: https://doi.org/10.5194/acp-2018-885).**
**In my opinion the authors decision that LWP is not meteorology needs to be properly justified or changed. Alternatively, the context of the results should be rephrased.**

> *We would like to thank the reviewer for showing that we should clarify why we separate meteorology and LWP together in order to best evaluate a sensitivity at the end of our results.*
>
> > *Environmental stability and entrainment directly affect the LWP so these parameters are not independent. In what follows, however, we consider the LWP separately from the local meteorology to separately evaluate two aspects of the indirect effect formulation. Since Twomey's original hypothesis of the aerosol indirect effect was based on holding LWP constant, we first examine the impact of increasing stringent constraints on LWP. Constraining LWP diminishes the effects of aerosol on cloud LWP itself allowing the sensitivity of the warm cloud CRE to aerosol to be isolated (Gryspeerdt et al.). More recently, numerous others have extensively demonstrated that aerosol indirect effects can be buffered by other environmental conditions. Since EIS and RH have been frequently adopted as proxies for these buffering effects, we further examine the impact of increasingly stringent constraints on these environmental characteristics. Our separation of 'cloud regimes' and 'meteorological regimes' is made only to contrast the magnitudes of their effects and does not imply that LWP is independent of EIS or RH. Ultimately it will be shown that all three factors must be accounted*

*together to adequately constraint the warm cloud radiative sensitivity to aerosol.*

*To replace:*
*"we consider the LWP separately from the local meteorology as it represents the cloud thermodynamics more than the local environmental conditions."*

*Further I have added to this section:*

*LWP responds to the humidity of the free atmosphere and inversion strength (De Roode et al. 2014). It has been shown the free atmospheric relative humidity can increase the sedimentation rate at the top of the cloud, altering the distribution of liquid throughout the cloud's vertical profile (Ackerman et al. 2004). Final results have constraints on LWP, EIS, and RH to account for relationships between meteorology and LWP.*

*We also added further acknowledgement that LWP is intrinsically tied to the meteorology, and only for clarity of terms within our paper we are addressing it separately from RH and EIS.*

*For the sake of clarity, we consider the LWP separately from RH and EIS, but we acknowledge that LWP is directly affected by the meteorology of the boundary layer.*

**The authors response to how CF is calculated seems not to answer my question.**
**The CF depends on the chosen area: if one would calculate the CF over a region of 4x4 km, while the typical clouds in the broader regions are much larger (say each cloud is 20x20km), the CF in the sub regions would tend to be mostly 0 or 1. If one would take the CF over 100x100km instead for the same broader region the total CF may be something else. One should be careful because when correlating the CF to other parameters it matter whether the CF is 1, 0 or in between (all CF may be correct, depending on the size of the averaged region). The authors choose segments of 12km which I think is too small to represent warm boundary layer cloud sizes. Perhaps showing the sensitivity of that choice (12 km segment) to the calculated CF would be enough.**

*Our cloud fraction is an along track cloud fraction. The width is always fixed at 1km, while we set the along track length at 12 km. Cloud fraction is found on a pixel by pixel basis, meaning at pixel i the cloud fraction is found from (i-5) to (i+6). At pixel i+1, the cloud fraction is found from (i-4) to (i+7). The track is not subdivided into 12 km lengths.*

[Figure]

*We chose a smaller scale like 12km to better represent smaller scale changes in cloud fraction that occurs in trade cumulus or during the stratocumulus to cumulus transition. Smaller changes to the clouds could be scaled out at higher resolutions like 48 or 95 km cloud fractions. As you can see from the above chart of ln(AI) against cloud fraction at different scales, as the scale increases from 12km to 96km, the slope decreases. Even a slight decrease in slope means that some changes in cloud fraction are being minimized by scale. Our choice of scale is to represent even small changes in the cloud extent, rather than derive an average cloud size. Our linear regressions are focused on the changes in cloud extent rather than absolute size. Choosing a small resolution captures small changes to cloud extent and is the best choice for this study. The processes that affect the cloud extent of warm clouds start with small-scale changes to the cloud size. In marine stratocumulus to cumulus transitions, the stratocumulus deck is gradually broken up into cumulus, however this would be lost with higher scales which would only capture a suddenly large change in cloud fraction instead of the gradual reduction represented on a smaller scale.*

**The authors say that the thinnest clouds are less sensitive to aerosols. This is in contrast to what one would expect: the albedo is not yet saturated for thin**

**clouds and therefore both Twomey and LWP effects would tend to increase the albedo more than in thicker clouds.**

**Unless having a plausible physical explanation for this result, I doubt the authors analysis. What can change the CRE and reduce the thin clouds sensitivity? Is it evaporation of the smaller drops? This would mean a negative sensitivity?**

> *It is important to distinguish between thin clouds and the very thinnest cloud LWP category in this analysis (0.02-0.04 kg/m2) that are extremely sensitive to entrainment-evaporation effects. This effect depends on the meteorological regime as discussed in the Discussion section and is especially prominent in drier environments.*
>
> *It may be true that in an idealized model, without effects of entrainment or evaporation, extremely thin clouds would exhibit the largest indirect effect as Twomey surmised. However real world clouds exist and interact with the environment. As the drop size decreases in these tenuous clouds (.02-.04 kg/m2), they become much more susceptible to evaporation. Evaporation in any part of the cloud reduces the overall cloud's radiative effect. In particular, the cloud edges would be extremely susceptible to evaporation, even without increased entrainment, as the cloud edges would be thinner and closer to drier air than the center.*
>
> *It is also important to note that the observed LWP is a scene wide average. The entire cloud itself is not a single, fixed depth. Therefore, changes in depth, like those that occur at the edges, or enhanced entrainment, stimulated by certain meteorological conditions, can lead to a reduced sensitivity to aerosol of the entire cloud.*
>
> *Finally, this could also be a result of semi-direct effects depending on the region. Thin clouds with embedded aerosol layers at the top will reduce the sensitivity by reducing the CRE. The reduced sensitivity for low LWP clouds shown in Figure 2 is part of the motivation of further constraining the sensitivity by meteorology and region in order to understand and diagnose variation in the sensitivity depending on the conditions. You cannot quantify the sensitivity with all the regions and environments lumped together even with constraints on LWP.*

**P9 L26 "The sensitivity is only calculated if there are 100 observations within the regime and the linear regression Pearson correlation coefficient is greater than .4"**

**Why 100? Why .4?**

> *We have added to this sentence "The sensitivity is only calculated if there are 100 observations within the regime to ensure an adequate number of observations to regress against, and the linear regression Pearson correlation coefficient is greater than .4 to ensure the slope is a good fit within each regime." to explain our reasoning behind the numbers.*

**The relationship between turbulence and activation efficiency is still unclear.**

*Turbulence and vertical velocity alter the structure of a cloud. Turbulence moves CCN, water vapor, and un-activated aerosol around in a cloud, increasing the chances of aerosol being activated and altering the supersaturation of a cloud. Vertical velocity plays many roles, including increasing the amount of water vapor in a cloud by cooling rising air adiabatically. Processes that increase turbulence and/or vertical velocity are of direct importance to aerosol activation, hence why aerosol activation parameterizations include vertical velocity as a direct input on aerosol activation efficiency.*

**P20 L24 "Surface winds were not included in analysis because the dependence of the warm cloud radiative response to aerosols depends most on LWP, RH, and stability, with only some regions showing a dependence on surface winds in our initial analysis."**

**Can you show these results?**

*Regionally, when we regress against CRE, the surface winds have the lowest average R value (as shown below) compared to our other regime variables. We*

[Figure]

*chose RH at 700 mb and EIS because they are both tied to the boundary layer inversion strength, cloud amount, and act as constraints on how in-cloud turbulence and entrainment can act to either invigorate or inhibit aerosol-cloud interactions. LWP was chosen because in the original work of Twomey and Albrecht, their models assumed a constant LWP. We are not saying that surface winds are not important to aerosol-cloud interactions, surface winds can play a large role in modulating the amount of sea spray transported into the cloud and in supporting coupled cloud amount, however for our study we chose based on our observations available to use RH at 700 and EIS instead for our 'meteorological regimes.'*

[revised manuscript text omitted]

To hold the cloud state fixed, the sensitivity is found for distinct seven LWP regimes (k) and summed to yield a mean sensitivity:

$$\lambda_{LWP} = \sum_{k=1}^{N_{LWP}} \left( -\frac{\partial CRE}{\partial \ln(AI)} \right)_k [..^8] W_k \tag{5}$$

Where $[..^9]W_k$ is $[..^{10}]$fraction of observations in cloud state k$[..^{11}]$:

$$W_k = \frac{\text{Number in Cloud State k}}{\text{Total Number}} \tag{6}$$

In our results, we evaluate the efficacy of increasing and decreasing the number of cloud states.

Similarly, the sensitivity within environmental regimes, defined by the estimated inversion strength and relative humidity of the free atmosphere, can be computed, weighted, and summed to account for meteorological covariability with ten regimes of each EIS (i) and $RH_{700}$ (j), where $[..^{12}]W_{i,j}$ is the $[..^{13}]$weighting factor for each environmental regime:

$$\lambda_{ENV} = \sum_{j=1}^{N_{RH}} \sum_{i=1}^{N_{EIS}} \left( -\frac{\partial CRE}{\partial \ln(AI)} \right)_{i,j} W_{i,j} \tag{7}$$

Where $W_{i,j}$ is the fraction of observations in environmental regime i,j:

$$[..^{14}]W_{i,j} = [..^{15}][..^{16}][..^{17}]\frac{\text{Number in Environmental Regime i,j}}{\text{Total Number}} \tag{8}$$
* * *
[9] removed: N

[10] removed: the number of observations of

[11] removed: .

[12] removed: N

[13] removed: number of observations within each environmental regime

By extension, both cloud and environmental conditions can be controlled via:

$$\lambda_{BOTH} = \sum_{k=1}^{N_{LWP}} \sum_{j=1}^{N_{RH}} \sum_{i=1}^{N_{EIS}} \left( -\frac{\partial \text{CRE}}{\partial \ln(\text{AI})} \right)_{i,j,k} [..^{18}] \text{W}_{i,j,k} \tag{9}$$

Where $[..^{19}]\text{W}_{i,j,k}$ is $[..^{20}]$ fraction of observations in both cloud state k and environmental regime i,j:

$$\text{W}_{i,j,k} = \frac{\text{Number in Environmental Regime i,j and Cloud State k}}{\text{Total Number}} \tag{10}$$

5   Finally, it is recognized that these bulk constraints do not fully capture all of the local factors that influence aerosol-cloud interactions. AI alone does not fully constrain the effect of aerosol composition which varies regionally. Thus, to control for these unaccounted for local effects, the sensitivity is further constrained by finding Eqn (9) on a 15° by 15° scale with four cloud state regimes (k), five regimes of stability (i), and five regimes of $RH_{700}$ (j) for each of the 152 regions (l).

$$\lambda_{ALL} = \sum_{l=1}^{N_{Reg}} \sum_{k=1}^{N_{LWP}} \sum_{j=1}^{N_{RH}} \sum_{i=1}^{N_{EIS}} \left( -\frac{\partial \text{CRE}}{\partial \ln(\text{AI})} \right)_{i,j,k,l} [..^{21}] \text{w}_{i,j,k,l} \tag{11}$$

10   Where $\text{W}_{i,j,k,l}$ is fraction of observations in region l in both cloud state k and environmental regime i,j.

**3   Results**

**3.1   Unconstrained Sensitivity**

The global sensitivity of warm cloud SW forcing to aerosol without any constraints described by Equation (4) is -12.81 $\frac{\text{W m}^{-2}}{\ln(\text{AI})}$ (Figure 1). This seems to capture the warm cloud AIE, after all the shortwave CRE increases with aerosol loading as expected.

15   However, this unconstrained estimate ignores the roles of buffering and covariance. The indicated variation of SW CRE within each ln(AI) bin alludes to variation in the overall effect not captured by a single linear regression. Although the $R^2$ is high, without constraints the increase in shortwave CRE cannot be attributed to only aerosol. Furthermore, from this estimate, no information is made known on how the sensitivity varies regionally, how cloud processes affect the AIE, or whether particular cloud states may be influenced more strongly by aerosol than others.
* * *
[19] removed: N

[20] removed: the number of observations within each environmental regime when constrained further by each of the cloud state regimes k .

[revised manuscript text omitted]